biochemistry/biophysics/plant science

non-photochemical quenching, photodamage, proton motive force, qE, photosynthetic control fund, unsupervised learning

**Author for correspondence:**
David M. Kramer
e-mail: kramerd8@msu.edu

# Light potentials of photosynthetic energy storage in the field: what limits the ability to use or dissipate rapidly increased light energy?

Atsuko Kanazawa[1,2], Abhijnan Chattopadhyay[1,3], Sebastian Kuhlgert[1], Hainite Tuitupou[1], Tapabrata Maiti[3] and David M. Kramer[1,4]

[1]MSU-DOE Plant Research Lab, [2]Department of Chemistry, [3]Department of Statistics and Probability, and [4]Biochemistry and Molecular Biology, Michigan State University, East Lansing, MI 48824, USA

AK, 0000-0001-9570-3419; SK, 0000-0002-0108-6921; DMK, 0000-0003-2181-6888

The responses of plant photosynthesis to rapid fluctuations in environmental conditions are critical for efficient conversion of light energy. These responses are not well-seen laboratory conditions and are difficult to probe in field environments. We demonstrate an open science approach to this problem that combines multifaceted measurements of photosynthesis and environmental conditions, and an unsupervised statistical clustering approach. In a selected set of data on mint (*Mentha* sp.), we show that 'light potentials' for linear electron flow and non-photochemical quenching (NPQ) upon rapid light increases are strongly suppressed in leaves previously exposed to low ambient photosynthetically active radiation (PAR) or low leaf temperatures, factors that can act both independently and cooperatively. Further analyses allowed us to test specific mechanisms. With decreasing leaf temperature or PAR, limitations to photosynthesis during high light fluctuations shifted from rapidly induced NPQ to photosynthetic control of electron flow at the cytochrome $b_6f$ complex. At low temperatures, high light induced lumen acidification, but did not induce NPQ, leading to accumulation of reduced electron transfer intermediates, probably inducing photodamage, revealing a potential target for improving the efficiency and robustness of

photosynthesis. We discuss the implications of the approach for open science efforts to understand and improve crop productivity.

# 1. Introduction

While oxygenic photosynthesis supplies energy to drive essentially all biology in our ecosystem, it involves highly energetic intermediates that can generate highly toxic reactive oxygen species (ROS) that can damage the organisms it powers [1]. Thus, the energy input into photosynthesis must be tightly regulated by photoprotective mechanisms that act at several key steps in the light reactions. The balance and kinetics of this regulation is an active target for crop improvement.

One class of photoprotective processes, known as non-photochemical quenching (NPQ), dissipates absorbed light energy as heat, thus diverting energy away from photosystem II (PSII) [2], decreasing the accumulation of reactive intermediates. This photoprotective capacity comes at the cost of decreased photochemical efficiency, and thus the organisms must regulate NPQ to balance the avoidance of photodamage with efficient energy conversion [3,4]. There are several forms of NPQ that differ in their mechanisms and rates of activation and deactivation. The most rapid NPQ form is qE, which is activated by acidification of the thylakoid lumen by the proton gradient ($\Delta$pH) component of the thylakoid proton motive force (*pmf*) [2]. Lumen acidification activates the violaxanthin de-epoxidase or VDE [5–8] resulting in the conversion of violaxanthin (Vx) to antheraxanthin (Ax) and zeaxanthin (Zx); and protonation of PsbS, an antenna-associated protein required for qE [2], which appear to act cooperatively in setting the extent of qE. The conversion of Vx to Ax and to Zx is typically much slower than the rapidly reversible protonation of PsbS [2], and during prolonged illumination, the responses of qE will probably be limited by the rate of acidification and de-acidification of the thylakoid lumen, which are, in turn, governed by ion movements in the chloroplasts [9–11]. Slower forms of NPQ have also been demonstrated [12], including qI, which is related to the photodamage and repair of photosystem II (PSII) or qZ, which related to the accumulation of Zx (independently from qE) [13], qH, related to cold and high light stress [13], and qT, related to antenna state transitions [14].

The acidification of the thylakoid lumen also controls electron transfer at the cytochrome $b_6f$ complex, a process called photosynthetic control (PCON) [15–20], which prevents the build-up of electrons on the acceptor side of photosystem I (PSI) that can lead to photodamage [15,21–23]. Interestingly, PCON and qE (both responses to lumen acidification) are expected to have opposing effects on $Q_A$ redox state. High levels of PCON in the absence of qE would lead to accumulation of plastoquinol ($PQH_2$) and the reduced form of the PSII electron acceptor, $Q_A$-, which can potentiate photodamage. Thus, these two processes must be tightly coordinated, with qE being activated at lumen pH somewhat less acidic than PCON [15].

Plants in natural environments are exposed to rapidly changing environmental conditions, especially light, which can change by orders of magnitude in less than a second. It has become clear that rapid and unpredictable fluctuations in light intensity can be more damaging than more gradual changes [22,24–32]. This sensitivity can partly be related to the build-up of reactive redox intermediates and thylakoid *pmf*, which can occur following low-to-high light transitions much more rapidly than the activation of photoprotective NPQ and PCON, leaving the photosynthetic apparatus prone to photodamage. Also, the slow recovery of NPQ following a decrease in light intensity can lead to substantial losses of photosynthetic efficiency [33].

Recently, it has been reported that engineering plants with increased expression levels of VDE and zeaxanthin epoxidase (ZE), resulted in accelerated formation and reversal of qE accompanied by increased plant productivity [3], suggesting that it may be possible to increase yield in crops by modifying photosynthetic regulatory responses.

On the other hand, we lack comprehensive surveys of the range of natural response of photosynthesis to real environmental fluctuations, in part because of a lack of deployable scientific equipment and methods to probe these processes in the field. Consequently, it has not been possible to assess the mechanistic bases of extant natural variations in these processes, their possible benefits or trade-offs, or which of these may be most useful for crop improvement.

Here, we introduce a method and proof-of-concept field data results to address the following questions: Can we assess the extent of natural variations in rapid responses to fluctuations in photosynthetically active radiation (PAR) intensity for both electron flow and photoprotection? How

do these limitations depend on environmental conditions? What are the mechanisms that underlie these variations in responses to rapidly fluctuating light in the field?

We define the term 'light potential' (LP) as the plant to respond to sudden increases in PAR, either by using it for productive photochemistry or up-regulating photoprotective mechanisms. In effect, LP is the response of photosynthesis to removal of light limitations. Here, we introduce an approach to both measure and analyse these variations in LP, focusing on one species, *Mentha* sp., under a limited set of conditions, and applied these to testing among a set of mechanisms for modulating that can be distinguished based on a range of optical measurements available using the MultispeQ 2.0 device, including: (i) PSI acceptor-side limitations to electron transfer; (ii) increased NPQ, which limits the input of light energy into photosystem II (PSII); and (iii) PCON, in which acidification of the lumen slows electron transfer at the level of plastoquinol ($PQH_2$) oxidation by the cytochrome $b_6f$ complex.

The results show that the approach can effectively be used to assess the range of variations in LP under field conditions, as well as to test specific hypothetical models, setting up a broad-scale, multiple-participant, open science approach to exploring the responses across multiple species, genotypes and environments. The results also reveal, at least in *Mentha*, unexpected leaf temperature-dependent limitations in the rapid formation of NPQ that result in the accumulation of reduced PSII electron acceptor, $Q_A$, and a high thylakoid *pmf*, conditions likely to promote the formation of ROS.

# 2. Material and methods

## 2.1. Plants and leaf sampling

Measurements were made in a population of *Mentha spicata* (Spearmint) plants that have been maintained at the MSU Horticulture Gardens (East Lansing, MI, USA) for at least 10 years. The GPS locations of all measurements are included in the online dataset (https://photosynq.org/projects/ rapid-ps-responses-pam-ecst-npqt-mint-dmk). Although it was not practical to exhaustively capture the lifecycle of the plants, the experimental strategy sampled a sufficiently wide range of conditions to allow clear patterns to emerge in the relationships between response behaviours and environmental parameters, as described below. The experiment took place over a 9-day experimental window between 21 July and 2 August 2019 (electronic supplementary material, figure S1A), sampling a range of times of day (between 5.30 and 18.40 local time), temperatures, etc. (electronic supplementary material, figure S1B). Measurements were made at multiple, alternating canopy levels and positions (subjectively at high, middle and low canopy levels) from early morning to later afternoon (electronic supplementary material, figure S1B), and at multiple locations across the plots on each day. Plants were up to 1 m in height. The leaf area index was not measured in this experiment. For the weather data, see the MSU Enviroweather website (https://mawn.geo.msu.edu/station.asp?id=msu, MSU Horticulture Teaching & Research Center, latitude: 42.6734, longitude: −84.4870, elevation: 264 m).

## 2.2. Measurements of photosynthetic and related parameters using MultispeQ 2.0

Optical measurements were made using MultispeQ 2.0 hand-held instruments (https://photosynq.com), based on that presented by Kuhlgert *et al.* [34] and calibrated using the CaliQ calibration system (https:// photosynq.com/caliq). The LP protocol used in the experiments can be found in the online project information (rapid-ps-responses-with-ecs-fast-ecs-dirk-and-npqt-dmk) as illustrated in figure 1. The protocol was designed to strike a balance among the needs for sampling large numbers of leaves, the desire for detailed spectroscopic measurements and the length of time the plant could be exposed to increased or decreased PAR. The full protocol, with measurements at ambient, after 10 s full sunlight and 10 s dark required about 35–40 s, at the limit of the time scale over which most researchers could steadily clamp a leaf in the instrument. The implications of the 10 s illumination and recovery time are discussed in the Results and Discussion sections.

In the first stage of the protocol (figure 1*a*), the MultispeQ was programmed to continuously (at about 5 Hz) measure PAR and reproduced these levels using a red actinic LED (655 nm emission peak) illuminating the adaxial surface of the leaf. When the MultispeQ detected that a leaf was clamped in the chamber, a series of measurement sequences was initiated. After a few seconds of illumination at ambient PAR ($PAR_{amb}$) to allow for settling and setting of gains, the first set of measurements was made, estimating at $PAR_{amb}$ linear electron flow (LEF) ($LEF_{amb}$), NPQt ($NPQ_{amb}$) and other photosynthetic parameters (figure 1*b*).

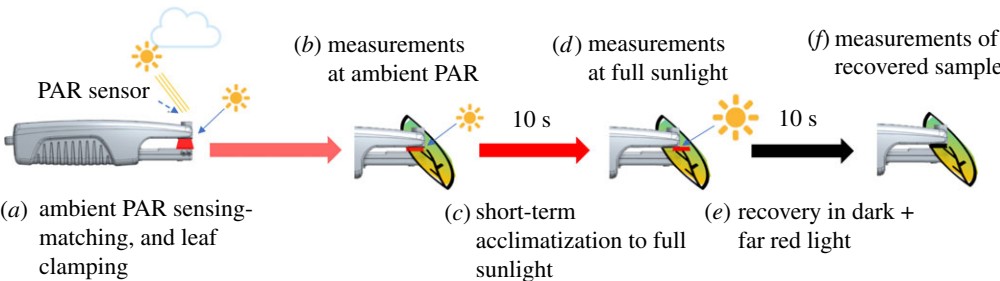

**Figure 1.** Experimental procedure of NPQ light potential designed to detect the change in NPQ induced under different light intensities. (*a*) A sensor on the MultispeQ continuously detects the ambient PAR in the field and reproduces this PAR value using an internal LED. (*b*) When the leaf clamp is closed over a leaf, the experiment begins by recording the local PAR, leaf temperature, ambient temperature, leaf angle and GPS position. After a short period of illumination at the measured ambient PAR, the first set of optical measurements are recorded. (*c*) Once completed, the leaf is exposed to a period of high PAR (2000 µmol m$^{-2}$ s$^{-1}$ equivalent to full sunlight) for 10 s. (*d*) The optical measurements are repeated at high PAR. (*e*) The leaf is then dark adapted (actinic light is switched off), with weak far-red background light for 10 s. (*f*) A final set of optical measurements is made to assess rapid dissipation of NPQ and reoxidation of accumulated reduced intermediates. Each set of optical measurements includes chlorophyll fluorescence and absorbance changes to give estimates of $\Phi_{II}$, LEF, NPQt, qL (Q$_A$ redox state); ECSt, P$_{700}$ redox state and g$_{H+}$ (relative proton conductivity of the thylakoid ATP synthase), as described in Material and methods. Measurements taken at ambient and high light are designated with the subscripts amb and high, as in LEF$_{amb}$, NPQ$_{amb}$, qL$_{amb}$ and LEF$_{high}$, NPQ$_{high}$, qL$_{high}$.

The actinic light was then increased to approximately full sunlight (2000 µmol m$^{-2}$ s$^{-1}$ red light) for 10 s (figure 1*c*), after which the photosynthetic measurements were repeated (figure 1*d*), yielding measurements of LEF$_{high}$, NPQ$_{high}$, etc. We chose full sunlight, rather than an artificially intense super-saturation light, to estimate LPs that could occur in the field, and not the absolute maximum, and to avoid non-physiological or photoinhibitory effects. Thus, the LPs of various processes will be limited as PAR$_{amb}$ approaches full sunlight.

In the third stage of the experiment, the actinic light was then switched off, and a weak far-red light switched on for 10 s (figure 1*e*), following another repetition of the measurements to assess the extent of NPQ$_t$ after relaxation (NPQ$_{rec}$, figure 1*f*). Environmental parameters including PAR, temperature, humidity, leaf temperature, leaf angle and GPS location were measured either prior to or following the physiological measurements.

Chlorophyll fluorescence changes were measured using MultispeQ 2.0 devices to estimate PSII quantum efficiency ($\Phi_{II}$) and linear electron flow (LEF) [35,36], as well as qL, a measure of the fraction of Q$_A$ in the oxidized state [37], and the extent of NPQ based on the rapid 'total' NPQ method developed by Tietz *et al.* [38], designated as NPQ$_t$. Just prior to the saturation pulses, dark interval relaxation kinetics (DIRK, dark interval of approx. 300 ms) of the absorbance changes around 520 nm attributed to the electrochromic shift (ECS) were recorded. Fitting the ECS signals to exponential decay curves yielded estimates of the relative light–dark differences in thylakoid *pmf* (ECS$_t$) and the proton conductivity of the chloroplast ATP synthase ($g_H+$), as described in [16,39,40]. To account for differences in leaf thickness, light path or number of chloroplasts in various leaves, the ECS$_t$ values were normalized to the relative chlorophyll contents as estimated by the SPAD parameter [34], which was measured at the end of the experiment. The extent of oxidation of P$_{700}$ in the light was estimated by the DIRK of infrared LED light using an LED measuring pulse with peak emission at approximately 810 nm.

Two features of the instrument's leaf clamp provide free air flow to the outside, preventing the depletion of CO$_2$ during the measurements. First, there is an approximately 3 mm space (or gap) between the leaf surface and the light guides, allowing lateral air flow. Second, this gap is connected to the external environment via a pair of rectangular (2 × 3 mm) air flow guides on the sides of the light guide, leading to the instrument case, the rear of which is open to the air. We thus do not expect any significant restriction of CO$_2$ diffusion to the leaf surface. To check this, we compared measurements of 10 separate mint leaves made with an unmodified instrument with 10 made with an additional air pump that provided approximately 300 ml min$^{-1}$ air exchange in the leaf clamp. We observed no significant differences ($p > 0.5$) in LEF values measured as above, under ambient (LEF$_{amb}$) or high (LEF$_{high}$, 2000 µmol m$^{-2}$ s$^{-1}$) red light, indicating that the air flow in the device was sufficient to prevent substantial CO$_2$ restriction to the leaves.

## 2.3. Environmental conditions during light potential measurements in the field

Electronic supplementary material, figure S1A–C shows the distributions of environmental factors (light intensities, leaf temperatures) for the measurements analysed in this study. The MultispeQ sensor was positioned by the user to be parallel to the leaf surface, so that the cosine-corrected PAR sensor should effectively estimate PAR absorbed by the leaf surfaces *in situ* throughout their canopy, and thus the ambient PAR ($PAR_{amb}$) values were dependent on both time of day (diurnal cycle, electronic supplementary material, figure S1B) and by leaf angle (electronic supplementary material, figure S1C). Ambient temperature and leaf temperatures ($T_{leaf}$) were dependent on time of day, with obvious influences from weather-related fluctuations (electronic supplementary material, figure S1A, B). We chose to compare results with $T_{leaf}$, rather than ambient temperature, to better reflect the effects on leaf photosynthetic processes. We note that previous results, e.g. Kuhlgert *et al.* [34], indicate that there may also be significant interactions between canopy position and photosynthetic parameters, though the current experiment did not explicitly record these positions, but rather sampled them as described in Material and methods.

## 2.4. Data calculations and cleaning

Data from the PhotosynQ platform were reprocessed and cleaned to improve the estimation of decay constants for ECS and near-infrared absorbance changes. As with any field experiments, some results were found to have obvious errors or be out of acceptable ranges, and were removed from the analysis. However, all original data were maintained in the online platform, allowing the reader to explore and reanalyse the effects of our data cleaning procedures. The rules and code for data flagging are defined in the Jupyter Notebook (see electronic supplementary material 'Data Cleaning Notebook'). A total of 292 points were flagged from a total of 1346 original measurements. The majority of the flagged measurements (179) were due to a defective device. The remaining 113 flagged points can be attributed to user error (e.g. leaf movements during measurements) or poor signal-to-noise that resulted in parameter values outside the theoretical ranges.

# 3. Results

## 3.1. Field measurements of photosynthetic parameters under ambient and rapidly elevated PAR

Figure 2*a* shows LEF measured at $PAR_{amb}$ ($LEF_{amb}$) plotted against ambient $PAR_{amb}$ and leaf temperature ($T_{leaf}$, see coloration of points). The plots use the square root of PAR to better resolve the results at lower $PAR_{amb}$, and to partially linearize the responses. $LEF_{amb}$ increased with increasing $PAR_{amb}$, with a roughly hyperbolic dependence and an apparent half-saturation point of about 350 µmol photons $m^{-2} s^{-1}$, reaching maximum values of about 250 µmol electrons $m^{-2} s^{-1}$ at 1700 µmol photons $m^{-2} s^{-1}$.

Upon 10 s of exposure to 2000 µmol photons $m^{-2} s^{-1}$ increased LEF to generally higher values ($LEF_{high}$ figure 2*b*), indicating that $LEF_{amb}$ was at least partly light-limited under all of the conditions. Note that each $LEF_{amb}$ point was taken on different leaves at different times (Material and methods) and has corresponding $LEF_{high}$ and $LEF_{high-amb}$ measurements. The relationship between measurements is illustrated in electronic supplementary material, figure S2, which shows selected pairs of $LEF_{amb}$ and $LEF_{high}$ connected by vertical line segments. The extent of $LEF_{high}$ was not uniform, but appeared to be strongly suppressed at low $PAR_{amb}$ and/or low $T_{leaf}$. The high light-induced difference in LEF ($LEF_{high-amb}$) increased with $PAR_{amb}$ at low light, reaching a peak at about 200 µmol photons $m^{-2} s^{-1}$, above which it declined as $PAR_{amb}$ approached $PAR_{high}$ and $LEF_{high}$ became light-saturated. The suppression of $LEF_{high}$ was due to large decreases in the quantum efficiencies of PSII (Phi2, figure 2*d*). Phi2 at $PAR_{amb}$ ($Phi2_{amb}$) were highest at low $PAR_{amb}$, and progressively saturated as light was increased. The opposite behaviour was seen with Phi2 measured after 10 s of high light ($Phi2_{high}$ figure 2*d*, grey symbols) which was lowest at low $PAR_{amb}$, and gradually increased with $PAR_{amb}$.

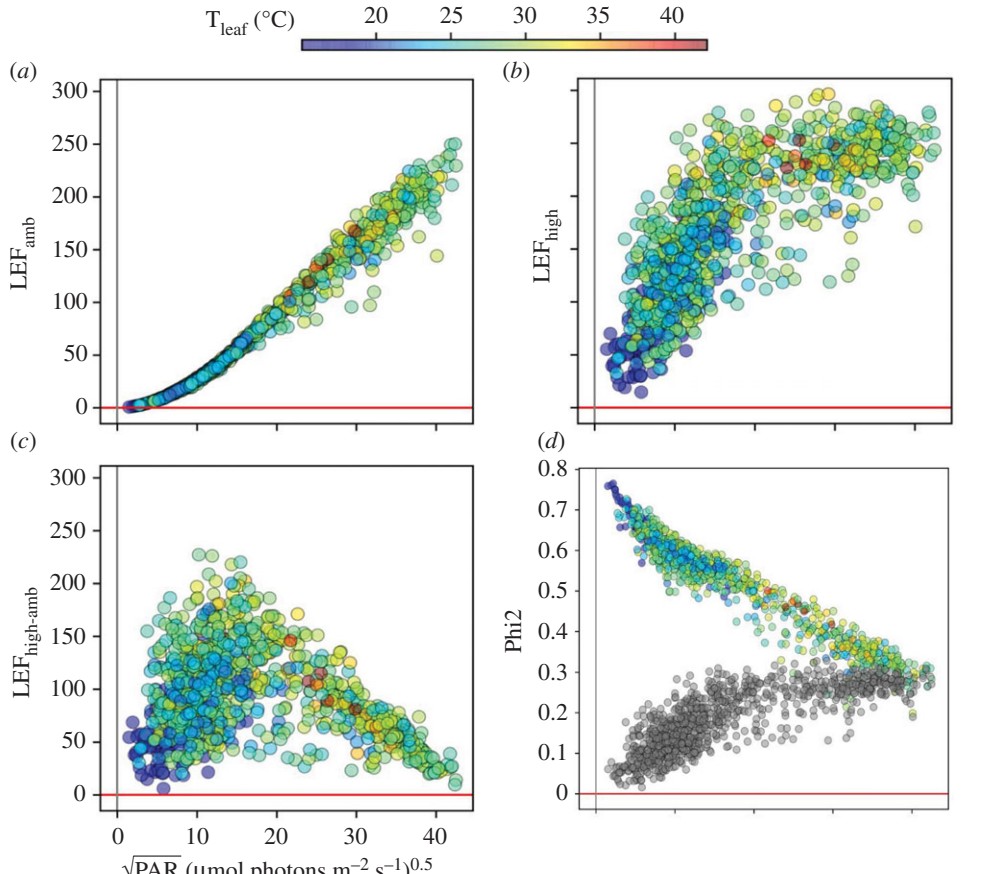

**Figure 2.** Light and temperature effects on LEF and photosystem II quantum efficiency ($\Phi_{II}$). Each parameter was plotted as a function of the square root of the ambient photosynthetically active radiation (PAR$_{amb}$, X-axis) and leaf temperature (T$_{leaf}$, coloration of points). (a) Dependencies of LEF measured at PAR$_{amb}$; (b) LEF measured at 10 s high light (LEF$_{high}$); (c) the high light-induced differences in LEF (LEF$_{high-amb}$); (d) the PSII quantum efficiencies measured under ambient PAR (Phi2$_{amb}$, points coloured by T$_{leaf}$) and at 10 s high light (Phi2$_{high}$, grey points).

## 3.2. Gaussian mixture model clustering analysis of field data

A simple linear effects model applied over the entire dataset (electronic supplementary material, table S1A) indicated strong correlations between LEF$_{amb}$ and both PAR$_{amb}$ and T$_{leaf}$, suggesting that both environmental factors controlled LEF$_{amb}$. However, such correlations may be coincidental since PAR and T$_{leaf}$ are both expected to be dependent on weather or time of day, as is clear from the strong statistical correlations between PAR and T$_{leaf}$. Also, the effects are likely to be co-dependent. For example, at low PAR$_{amb}$, LEF$_{amb}$ should be light-limited, and thus have minimal dependence on T$_{leaf}$, but at higher PAR$_{amb}$, may be more strongly controlled by temperature-dependent processes.

One approach to disentangling these effects would be to slice the data into segments, e.g. at different ranges of PAR$_{amb}$, and test for correlations with T$_{leaf}$ within each segment. However, arbitrarily chosen ranges for the segments can add bias, or fail to detect more complex interactions. We thus applied a Gaussian mixture model (GMM) clustering approach based on those presented earlier [41,42]. Because GMM is an unsupervised machine learning method, it can reduce bias in the selection of clusters that represent regions of distinct interactions among environmental and photosynthetic parameters. GMM assumes that the data points from the population of interest are being drawn from a combination (or mixture) of Gaussian distributions with certain parameters, and performs an optimization scheme to a sum of K Gaussian distributions, allowing for a completely unsupervised process, avoiding potential user bias. An expectation–maximization (EM) algorithm was used to fit the GMM to the dataset, generating a series of Gaussian components (clusters) with distributions characterized by specific means and covariance matrices. The optimal number of clusters was determined using the Bayesian information criterion (BIC), the value of the maximized log likelihood, with a penalty on the number of parameters in the model [41–44]. This approach also allows comparison of models with differing

parametrizations and/or differing numbers of clusters, because the volumes, shapes and orientations of the covariances can be constrained to those described by defined models [41].

Clusters obtained through GMM have both within cluster (intracluster) and between cluster (intercluster) variations. In order to test for intercluster variation, we used the clustering assignment obtained for one parameter (or response) and applied it on another response. Here we want to investigate what would be the distinctive behaviour of different responses if we have used the same configuration. Using the same set of cluster assignments to different responses, one might be skeptical of the clustering behaviour as responses interact differently with $PAR_{amb}$ and $T_{leaf}$. In that case, we might not be able to directly compare the intercluster behaviours of responses. To mitigate this issue, we use the GMM clustering as a tool to create a 'baseline' clustering configuration for one response and use that configuration over other responses. We set up our hypothesis as two responses are similar under the same configuration against they are not. If the interaction pattern of one response with $PAR_{amb}$ and $T_{leaf}$ changes over the other response, we reject our hypothesis and imply that different configurations of $PAR_{amb}$ and $T_{leaf}$ interact differently with responses. By doing this we are able to disentangle the effect of $PAR_{amb}$ and $T_{leaf}$ and infer regarding the intracluster variations as to be a key element to determine variations in the interactions between parameters and variations in environmental conditions, e.g. to assess if a relationship is modulated in different ways under different ranges of conditions. Also, as will be seen in the Discussion, intercluster variations (differences in the mean and covariances between clusters) can be used to differentiate distinct patterns of behaviour, or mechanistic interactions, between conditions.

As shown in electronic supplementary material, figure S3, GMM analysis of $LEF_{amb}$, $PAR_{amb}$ and $T_{leaf}$, found six distinct, compact clusters that differed in the mode of interaction among the photosynthetic and environmental parameters. Encompassing points with lower $PAR_{amb}$ showed strong (clusters 1, 2, 4 and 5) dependence of $LEF_{amb}$ on $PAR_{amb}$, with little contributions from $T_{leaf}$. By contrast, two clusters (3 and 6), which included points at higher $PAR_{amb}$, showed substantial dependencies on both $PAR_{amb}$ and $T_{leaf}$. These results are consistent with LEF being predominantly light-limited at low ambient PAR, but increasingly limited by temperature-dependent processes at higher PAR. The presence of these two classes of clusters indicates that $PAR_{amb}$ and $T_{leaf}$ are likely to affect $LEF_{amb}$ in independent ways. The fact that the shapes of the clusters were not determined with individual slicing under the individual parameters for $PAR_{amb}$ and $T_{leaf}$, but with a co-dependence on both $PAR_{amb}$ and $T_{leaf}$, suggests that, under some conditions, these effects interact, e.g. $T_{leaf}$ may affect the dependence of $LEF_{amb}$ on $PAR_{amb}$.

GMM identified five distinct clusters for interactions among $LEF_{high}$, $PAR_{amb}$ and $T_{leaf}$ (electronic supplementary material, figure S4). In contrast to the results on $LEF_{amb}$, clusters at lower $PAR_{amb}$ (1, 2 and 4) showed $LEF_{high}$ dependencies on both $T_{leaf}$ and $PAR_{amb}$, while cluster 3 showed correlations with $T_{leaf}$, but not with $PAR_{amb}$. The stronger dependence on $T_{leaf}$ of $LEF_{high}$ compared with $LEF_{amb}$ implies that the exposure to high light revealed additional rate limitations in $LEF_{high}$ that were more strongly controlled by both $T_{leaf}$ and $PAR_{amb}$ and that, at least under some conditions, these effects were independent of each other.

## 3.3. Analysis of NPQ

$NPQ_t$ measured under $PAR_{amb}$ ($NPQ_{amb}$, figure 3a) showed a positive correlation to $PAR_{amb}$, with an apparent tendency for smaller values at lower $T_{leaf}$. $NPQ_{amb}$ showed considerable variations, compared with $LEF_{amb}$, even at low $PAR_{amb}$, consistent with the idea that NPQ is governed not only by PAR but by metabolic, developmental or other environmental parameters.

Figure 3b shows $NPQ_t$ values measured at 10 s full sunlight ($NPQ_{high}$). The NPQ LP, or light-induced differences in NPQ ($NPQ_{high-amb}$) are shown in figure 3c. While $NPQ_{high-amb}$ was always positive, both $NPQ_{high-amb}$ and $NPQ_{high}$ were suppressed at low $PAR_{amb}$ or $T_{leaf}$. $NPQ_t$ measured after the 10 s dark recovery period ($NPQ_{rec}$, figure 3f) was consistently lower than $NPQ_{amb}$ and $NPQ_{high}$. The difference between $NPQ_{amb}$ and $NPQ_{rec}$ ($NPQ_{amb-rec}$, figure 3d) ranged from slightly negative at low $PAR_{amb}$, where the majority of $NPQ_{amb}$ was rapidly reversible, to about one at the higher $PAR_{amb}$, where about half of $NPQ_{amb}$ was rapidly reversed.

Overall, these results indicate that a large fraction (in many cases the majority) of $NPQ_{amb}$ as well as $NPQ_{high}$ recovered within 10 s of darkness and can probably be attributed to qE, and thus, under our conditions, qE is likely to be the most important form of NPQ for rapid adjustments to photoprotection. The residual, more slowly reversible, components reaching a little above 2 are likely to include qI or qZ [45,46], although the limited time frame for the protocol does not allow us to rule out contributions from longer-lived qE. It is also important to consider that the fraction of light

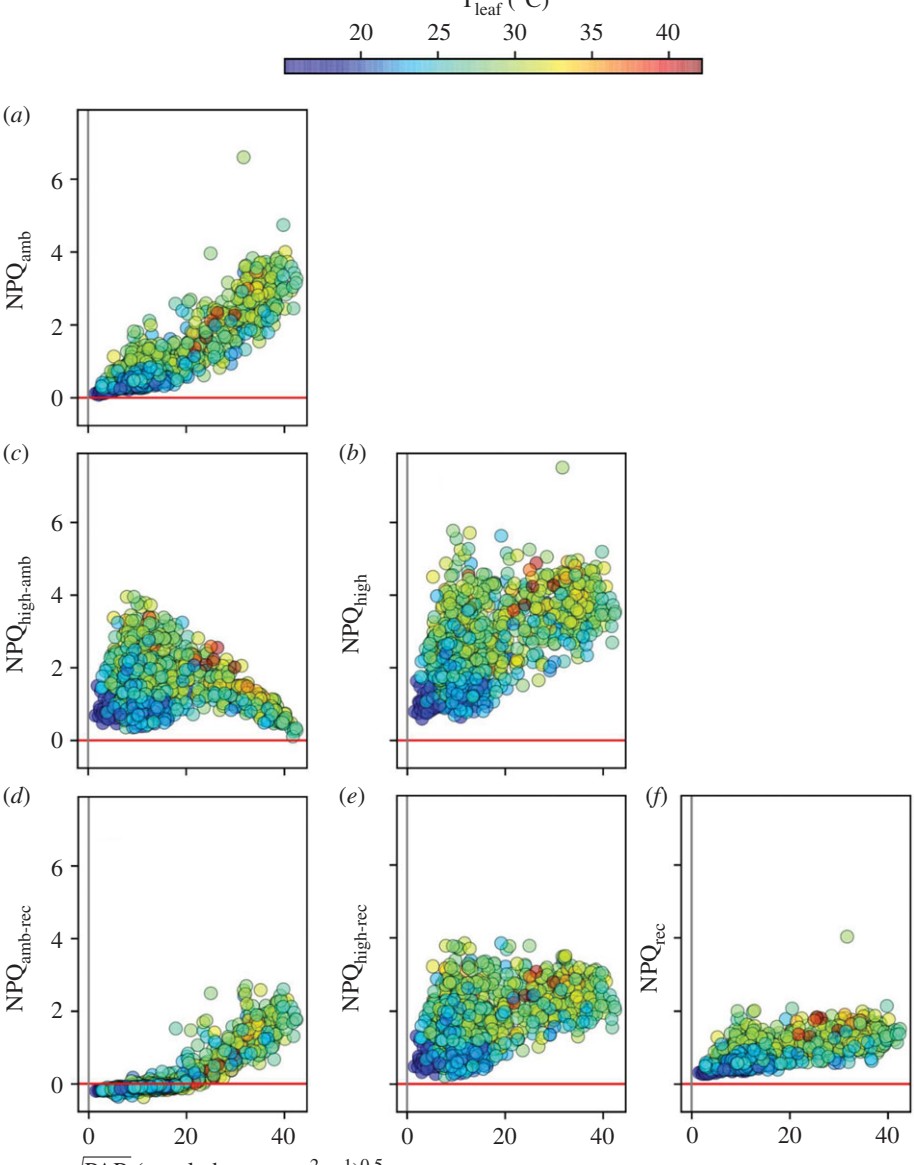

**Figure 3.** Light and temperature effects on NPQ. The NPQ parameter was plotted as functions of the square root of the ambient photosynthetically active radiation (PAR$_{amb}$, X-axis) and leaf temperature (T$_{leaf}$, coloration of points). (a) Induced NPQ measured at PAR$_{amb}$; (b) NPQ measured at 10 s high light (NPQ$_{high}$); (c) the high light-induced differences in NPQ (NPQ$_{high-amb}$); (d) the difference in induced NPQ level at ambient PAR and the 10 s recovery time in the dark (NPQ$_{amb-rec}$); (e) the difference in induced NPQ level at 10 s high PAR and the 10 s recovery time in the dark (NPQ$_{high-rec}$); (f) the NPQ level after 10 s in the dark (NPQ$_{rec}$).

energy dissipated by the NPQ, i.e. $\Phi_{NPQ}$, will also depend on the fraction of PSII in open states [37], which will also be impacted by ambient and fluctuating light, T$_{leaf}$ and other factors.

As with LEF, a simple linear effects model (electronic supplementary material, table S1B) showed strong interactions between T$_{leaf}$ and PAR$_{amb}$, on NPQ$_{amb}$, and the corresponding GMM analysis identified four clusters (electronic supplementary material, figure S5). Cluster 1, which encompassed the lowest range of PAR$_{amb}$ values, showed strong dependence on PAR$_{amb}$, with no significant dependence on T$_{leaf}$. The remaining clusters showed either dependence solely on T$_{leaf}$ (cluster 4) or co-dependence on PAR$_{amb}$ and T$_{leaf}$ (clusters 2 and 3). Because GMM clustering suggests that T$_{leaf}$ and PAR$_{amb}$ can interact or act independently, depending on conditions, we excluded the linear effects models and focused on GMM for analyses of the remaining parameters.

For the analysis of NPQ$_{high}$ (electronic supplementary material, figure S6), we used the clusters found for NPQ$_{amb}$ (electronic supplementary material, figure S5), allowing us to directly compare changes in correlations among parameters within each cluster [41]. Cluster 1, which encompassed the lowest range of PAR$_{amb}$ values, showed strong dependence of NPQ$_{high}$ on both PAR$_{amb}$ and T$_{leaf}$. This pattern of

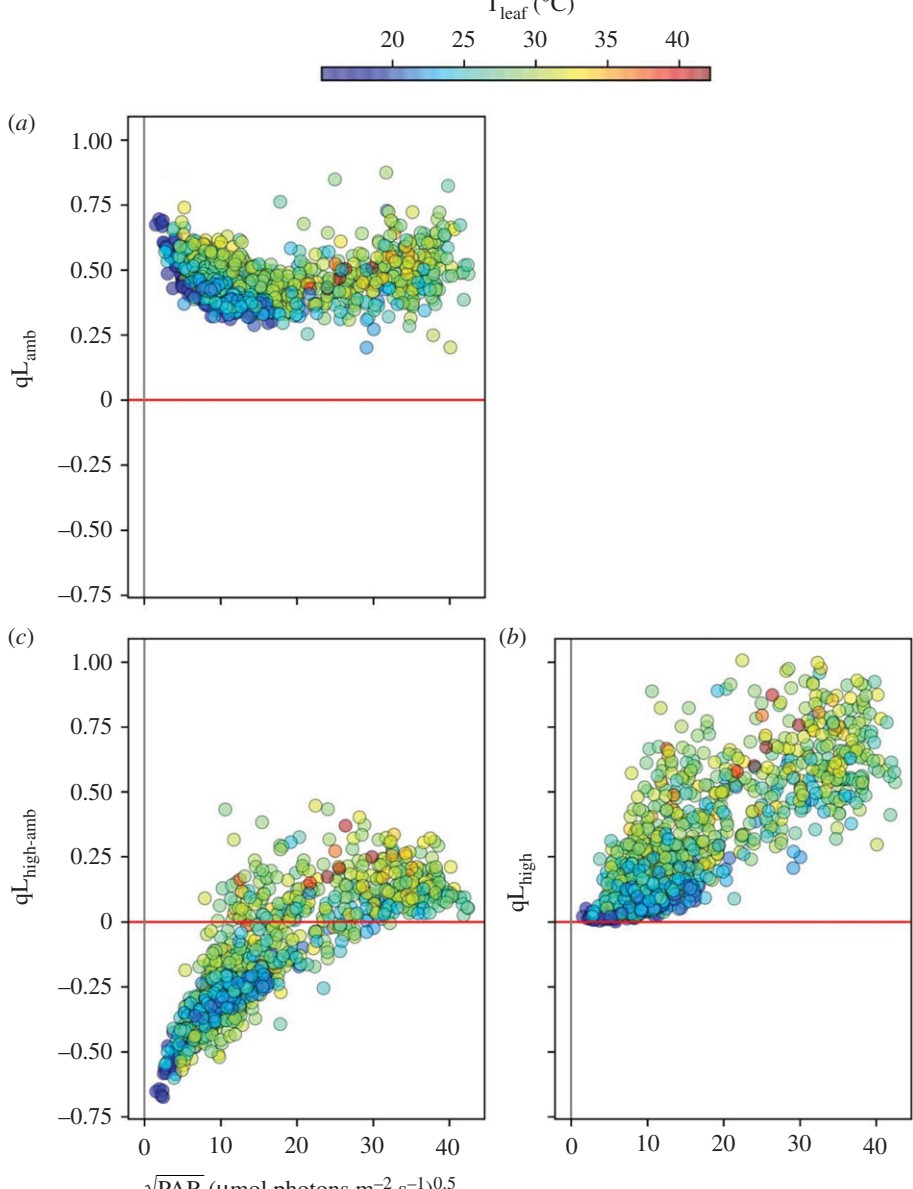

**Figure 4.** The light and temperature dependencies of the redox state of $Q_A$. The qL parameter, a measure of fraction of $Q_A$ in its oxidized state, was measured as described in Material and methods, (*a*) under ambient light ($qL_{amb}$), (*b*) at 10 s of high light ($qL_{high}$), and (*c*) the change in qL between high and ambient PAR ($qL_{high-amb}$) as a function of the square root of ambient PAR.

dependencies was in contrast to that for cluster 1 for $NPQ_{amb}$, which showed dependence solely on $PAR_{amb}$. At a higher range of $PAR_{amb}$ (cluster 3), $NPQ_{high}$ showed significant dependence solely on $T_{leaf}$, again in contrast to the corresponding cluster for $NPQ_{amb}$, which showed dependencies on both $PAR_{amb}$ and $T_{leaf}$. Overall, compared with $NPQ_{amb}$, $NPQ_{high}$ showed increased dependence on $T_{leaf}$ in all clusters, suggesting that it is more substantially controlled by metabolic or physiological factors (see Discussion).

## 3.4. The redox state of $Q_A$

Figure 4*a* shows the dependencies of $Q_A$ redox state (qL) on PAR and $T_{leaf}$. qL measured at $PAR_{amb}$ ($qL_{amb}$, figure 4*a*), was relatively constant (ranging from about 0.3 to 0.75) across $PAR_{amb}$, with somewhat higher values at both extremes of $PAR_{amb}$. Lower leaf temperatures appeared to be associated with lower qL values, over the entire range of $PAR_{amb}$, although the effect was particularly pronounced at low light. By contrast, qL measured at 10 s of high light ($qL_{high}$, figure 4*b*) showed strong dependence on $PAR_{amb}$, ranging from near zero (fully reduced $Q_A$) at low $PAR_{amb}$, to almost one (fully oxidized) at higher $PAR_{amb}$. Again, low $T_{leaf}$ appeared to correlate with lower $qL_{high}$

throughout the range of $PAR_{amb}$. Strikingly, as shown in figure 4$c$, the high light treatment induced two distinct effects: at low $PAR_{amb}$ and/or $T_{leaf}$, it induced a net reduction of $Q_A$, while it had the opposite effect at higher $PAR_{amb}$ and $T_{leaf}$.

GMM clustering for $qL_{amb}$, $PAR_{amb}$ and $T_{leaf}$ (electronic supplementary material, figure S7) identified four distinct clusters. In cluster 2, which encompasses points at low $PAR_{amb}$, significant associations were observed only between $qL_{amb}$ and $PAR_{amb}$. Clusters 1, 3 and 4 (at higher $PAR_{amb}$) showed co-dependencies between $qL_{amb}$ and both $PAR_{amb}$ and $T_{leaf}$. GMM clustering for $qL_{high}$, $PAR_{amb}$ and $T_{leaf}$ showed five distinct clusters (electronic supplementary material, figure S8). Clusters 1, 2 and 5, which encompassed generally lower ranges for $PAR_{amb}$ and $T_{leaf}$, showed $qL_{high}$ dependencies on both $PAR_{amb}$ and $T_{leaf}$. Clusters 3 and 4 (generally with higher $PAR_{amb}$ and $T_{leaf}$ values) showed only dependencies on $T_{leaf}$. The overall pattern of cluster behaviour was similar to that observed with respect to $NPQ_{amb}$ and $NPQ_{high}$.

## 3.5. P700 redox state

Figure 5 shows the extent of oxidized $P_{700}^+$ ($P^+$), based on the DIRK of absorbance changes at 810 nm. $P_{700}^+$ at $PAR_{amb}$ ($P^+_{amb}$, figure 5$a$), after 10 s of high light ($P^+_{high}$, figure 5$b$) and the light-induced difference ($P^+_{high-amb}$, figure 5$c$). The extent of $P^+_{amb}$ was nearly linearly related to $PAR_{amb}$. Increasing the light resulted in higher $P^+$ values ($P^+_{high}$), indicating that, in all cases, PSI became more oxidized at high light. The extent of the light-induced oxidation was dependent on $PAR_{amb}$, with lower extents at low $PAR_{amb}$, and a peak at about 200–300 µmol photons $m^{-2}\,s^{-1}$. The decrease at higher $PAR_{amb}$ was probably due to the accumulation of pre-oxidized $P_{700}$ prior to the high light treatment.

The full extent of $P^+_{high}$ was relatively constant over the conditions, suggesting that high light was able to nearly fully oxidize $P_{700}$. However, there was a slight trend to lower $P^+_{high}$ at the highest $PAR_{amb}$ or $T_{leaf}$, suggesting that total oxidizable PSI may have decreased at high light or temperatures, perhaps reflecting accumulation of PSI photodamage or electron sink limitations. Consistent with these general trends, GMM analyses of $P^+_{amb}$, $PAR_{amb}$ and $T_{leaf}$ identified four distinct clusters (electronic supplementary material, figure S9), with dependencies on either $PAR_{amb}$ by itself (clusters 3 and 4), or both $PAR_{amb}$ and $T_{leaf}$ (clusters 1 and 2). GMM clustering for $P^+_{high}$ identified five distinct clusters (electronic supplementary material, figure S10), that showed a positive dependency of $P^+_{high}$ on either $PAR_{amb}$ (cluster 1), or $T_{leaf}$ (cluster 5), or a small, negative dependence on $T_{leaf}$ (cluster 3).

## 3.6. ECSt and thylakoid *pmf*

Figure 6 shows dependencies of relative thylakoid *pmf*, estimated by normalized ECSt measurements, at ambient PAR ($ECSt_{amb}$, figure 6$a$) and after 10 s exposure to high light ($ECSt_{high}$, figure 6$b$). The high light-induced differences ($ECSt_{high-amb}$) are shown in figure 6$c$. $ECSt_{amb}$ showed strong, positive correlations with $PAR_{amb}$, similar to the responses of $NPQ_{amb}$ (figure 3$a$) and $P^+_{amb}$ (figure 5$a$). $ECSt_{high}$ values were, in general, larger than $ECSt_{amb}$, resulting in positive values for $ECSt_{high-amb}$. At low $PAR_{amb}$, $ECSt_{high}$ showed high variability, suggesting that the response is strongly dependent on other factors, but appeared to saturate (flatten) at higher $PAR_{amb}$. These behaviours were reflected in $ECSt_{high-amb}$, which showed strong variability at lower $PAR_{amb}$ or $T_{leaf}$, peaked at about 50–100 µmol photons $m^{-2}\,s^{-1}$, and saturated at higher $PAR_{amb}$.

GMM analysis of $ECSt_{amb}$ identified five distinct clusters (electronic supplementary material, figure S11). The cluster at the lowest range of $PAR_{amb}$ (cluster 1) showed dependence primarily on $PAR_{amb}$. The remaining clusters showed positive correlations between $ECSt_{amb}$ and $PAR_{amb}$, but negative correlations with $T_{leaf}$. By contrast, GMM of $ECSt_{high}$ (electronic supplementary material, figure S12) showed almost no dependence on either $PAR_{amb}$ or $T_{leaf}$, except at the lowest $PAR_{amb}$ (cluster 1) which showed negative correlations with $PAR_{amb}$ and positive correlations with $T_{leaf}$.

# 4. Discussion

## 4.1. Using PhotosynQ and MultispeQ to sample and resolve the effects of environmental fluctuations on photosynthetic processes

The MultispeQ measurements described above were designed to explore the photosynthetic responses of plants in a natural, fluctuating environment. In this type of field experiment, it is not possible to control

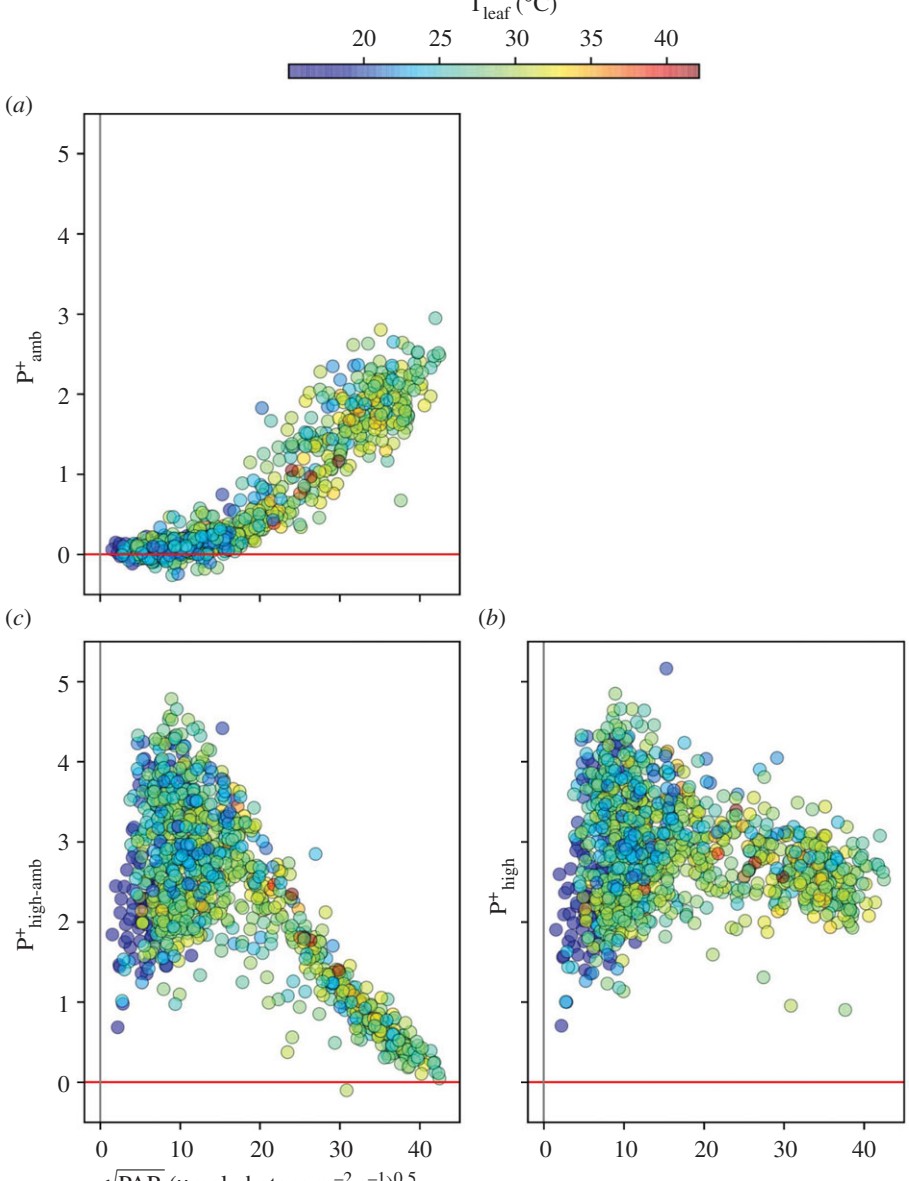

**Figure 5.** The light and temperature dependencies of the redox state of $P_{700}^+$. The redox state of P700 was measured using DIRK at 810 nm absorbance change (*a*) under ambient light ($P_{amb}^+$), (*b*) at 10 s of high light ($P_{high}^+$), and (*c*) the change in $P^+$ between high and ambient PAR ($P_{high-amb}^+$) as a functions of the square root of ambient PAR.

all variables. Rather, the strategy was to 'sample' responses under as many conditions as practical, while recording key metadata so that subsequent analyses can assess the impacts of various environmental fluctuations. Thus the observed trends may reflect both primary and acclimatory factors that change (or accumulate) over different time scales. Correlations that appear in such analyses can be used to test, at least to some extent, certain models, though it is important to note that more controlled experiments will be needed to fully determine cause–effect relationships, as discussed below.

A major outcome of the experiment is that, despite the fact that measurements were made over many plants, times, etc., clear patterns of responses emerged that allow us to make some broad conclusions about the responses of photosynthesis to ambient and rapidly changing light. For example, the majority of NPQ$_{high}$ was found, in general, to be rapidly reversible, suggesting that qE was the major contributor: at lower PAR$_{amb}$, that majority of NPQ$_{high}$ was rapidly induced (figure 3*c*), while at higher PAR$_{amb}$, pre-existing NPQ was rapidly recoverable (figure 3*e*).

Another important trend was the suppression of the LPs of both LEF (figure 2) and NPQ (figure 3) under some conditions, particularly under lower PAR$_{amb}$ and/or T$_{leaf}$. Further, strong decreases in LEF$_{high}$ were not always accompanied by compensatory increases in NPQ$_{high}$, implying that the

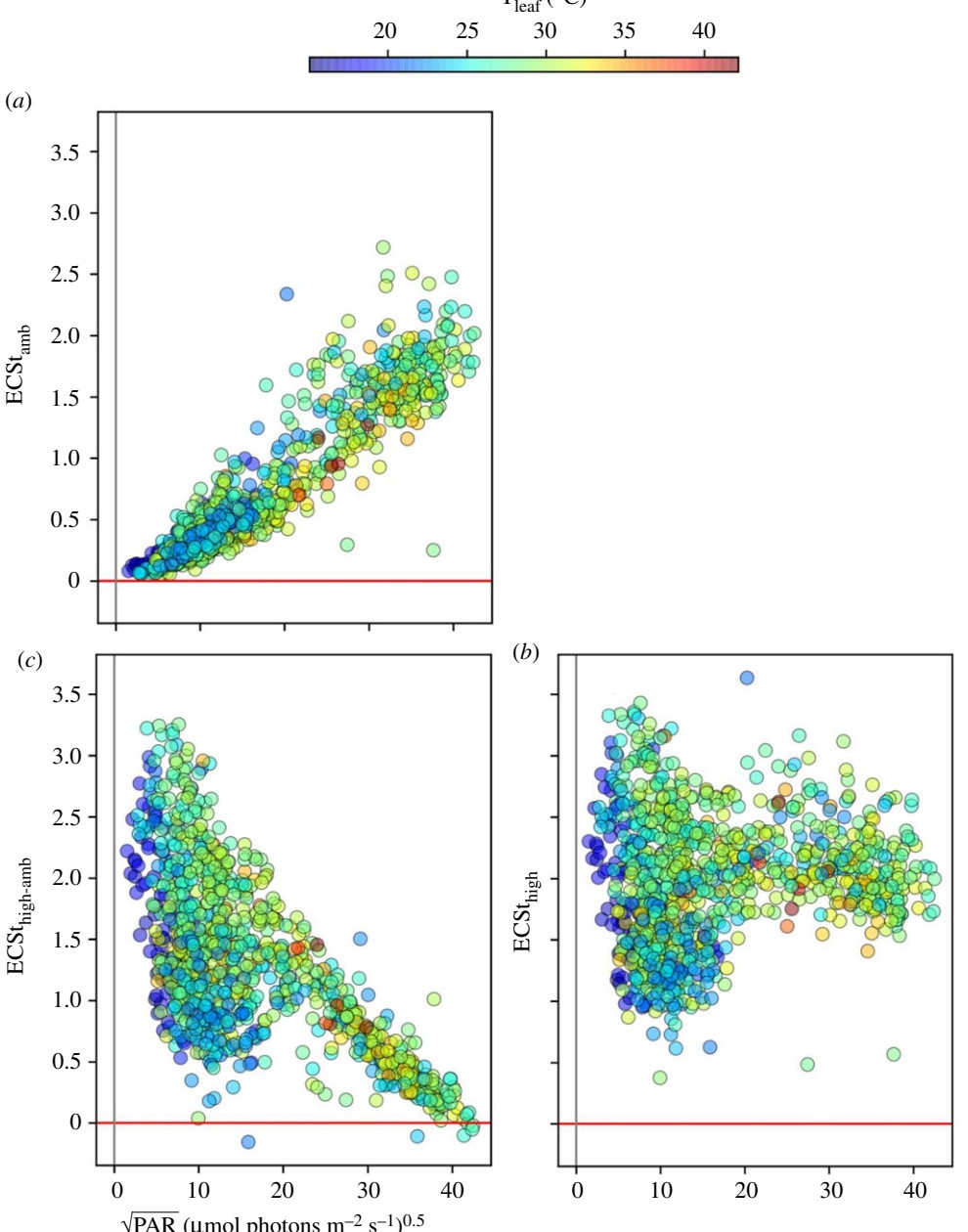

**Figure 6.** The light and temperature dependencies of the thylakoid *pmf* probed using ECSt signal. The *pmf* was measured using ECS (*a*) under ambient light (ECSt$_{amb}$), (*b*) at 10 s of high light (ECSt$_{high}$), and (*c*) the change in ECSt between high and ambient PAR (ECSt$_{high-amb}$) as a functions of the square root of ambient PAR.

productive and photoprotective LPs can be simultaneously suppressed under certain conditions, a situation that is likely to promote the formation of ROS and photodamage (see also below), with important implications for understanding the environmental robustness of photosynthesis [29].

## 4.2. Disentangling interacting environmental impacts on photosynthetic processes

A key challenge to the field experiment approach is in teasing apart effects from different environmental factors, especially considering that such factors may be co-dependent or interact with each other in complex ways. For example, in visual inspection, most of the parameters show apparent dependencies on both PAR$_{amb}$ and T$_{leaf}$ (e.g. figures 2–6) but, because increases in T$_{leaf}$ are often correlated with increases in PAR$_{amb}$, the effects of the two parameters may have been coincidental. It may also be that the environmental parameters interacted in complex ways, e.g. high PAR$_{amb}$ may have exacerbated the effects of low T$_{leaf}$. To address these issues, we applied an approach based on GMM to identify

clusters representing distinct interactions among parameters. The approach is unsupervised, thus eliminating potential bias, while allowing us to test for changes in the environmental dependencies among multiple environmental parameters (electronic supplementary material, figures S3–S12).

Analysis of GMM clusters implied that most parameters were dependent on both $PAR_{amb}$ and $T_{leaf}$, and at least under some conditions these effects are independent, or that one of the two factors predominates. Thus, the effects cannot be explained simply by coincidences between increased PAR and temperatures. Moreover, the non-rectilinear shapes of the clusters suggest that the effects of $PAR_{amb}$ and $T_{leaf}$ were interactive, e.g. changes in $T_{leaf}$ modulated the effects of $PAR_{amb}$ and vice versa. Overall, these interactions are in line with well-known temperature and PAR dependence of photosynthesis, but this type of analyses can reveal the specific combination of conditions that induce distinct behaviours, allowing for assessments of the involvement of specific mechanisms (see below) and to identify genotypic or management impacts on crop resilience and productivity.

At low $PAR_{amb}$, we expect steady-state photosynthesis to be predominantly light-limited, and thus the effects of $T_{leaf}$ should be low. As light increases, downstream biochemistry should become increasingly limiting. Because downstream energy storage and metabolic processes are likely to be more temperature dependent than photochemistry, this shift may allow us to distinguish between these types of limitations. Such behaviours are apparent in many of the measured parameters, e.g. $LEF_{amb}$, which was not substantially dependent on $T_{leaf}$ at low $PAR_{amb}$, but became co-dependent on $PAR_{amb}$ and $T_{leaf}$ at higher $PAR_{amb}$ (figure 2a; electronic supplementary material, figure S3), consistent with a progressive shift from light-limitation to assimilation-limitation. Similarly, $NPQ_{amb}$ was solely dependent on $PAR_{amb}$ in the cluster at low $PAR_{amb}$, but became increasingly dependent on $T_{leaf}$ as $PAR_{amb}$ increased (figure 3a). This shift is consistent with a control of $NPQ_{amb}$ by PAR (at low $PAR_{amb}$) and downstream metabolic processes, particularly at higher $PAR_{amb}$, e.g. due to regulation of the ATP synthase activity or cyclic electron flow (CEF) [47].

By contrast, $LEF_{high}$ and $NPQ_{high}$ showed much greater dependence on $T_{leaf}$, and these differences were more pronounced when the high light was imposed on leaves at low $PAR_{amb}$ and $T_{leaf}$, i.e. the opposite of what was seen for $LEF_{amb}$ and $NPQ_{amb}$. Interestingly, the $LEF_{high}$ rates achieved in leaves exposed to lower $PAR_{amb}$ were strongly suppressed below the maximum $LEF_{amb}$ values measured at higher $PAR_{amb}$ (compare figure 2a,b), This behaviour suggests that the suppression of $LEF_{high}$ occurs when abrupt increases in light overwhelm the activation of downstream energy storage and metabolic processes. This is generally consistent with observations that the activities of metabolic enzymes are regulated to match the availability of energy from the light reactions, which involve a large suite of co-regulatory processes, as extensively reviewed elsewhere, (e.g. [16,47–53]), but that these responses lag behind the changes in light. The *in situ* LP measurements afforded by MultispeQ show that these situations are very likely to occur under many field situations.

These results also imply that accurate estimates of LEF, NPQ and other photosynthetic parameters *under natural conditions* will require measurements under ambient light, because sudden changes in PAR can lead to severe perturbations in photosynthetic limitations or regulation. Attempts to 'simplify' field experiments by setting PAR to some constant value will lead to strong perturbations and the measured values will reflect these perturbations. The effects are vividly demonstrated by the opposite dependencies of $Phi2_{amb}$ and $Phi2_{high}$ on $PAR_{amb}$ (figure 2d), and validate the use of the PAR matching feature of the MultispeQ instrument. Nevertheless, as shown here, the effects of these perturbations can be informative, but care must be taken in extrapolating to the non-perturbed state. It is also important to keep in mind that the rates of acclimatization may vary substantially between species, and that these may be assessed by performing more intensive experiments with variable high light and dark recovery times.

## 4.3. Mechanisms for controlling the light potentials of LEF and NPQ using MultispeQ field data

The rapid reversal of $NPQ_{amb}$ and $NPQ_{high}$ over 10 s of dark indicated that, under our conditions, a large fraction of NPQ is in the form of qE (figure 3b,c), and thus dependent on lumen acidification and subsequent pH-dependent responses. It is important to note, though, that residual NPQ will contribute to decreases in photochemical efficiency, and that the extent of these effects will also be impacted by other factors, including the redox state of $Q_A$ [37]. Lumen acidification can be controlled by changes in proton influx (through changes in LEF and CEF), proton efflux through the ATP synthase and the partitioning of *pmf* into electric field ($\Delta\psi$) and $\Delta pH$ components, which in turn, are impacted by metabolic status, as proposed earlier [15,39]. Here, we explore the possible mechanistic bases for these effects, by comparing the correlations among MultispeQ measurements.

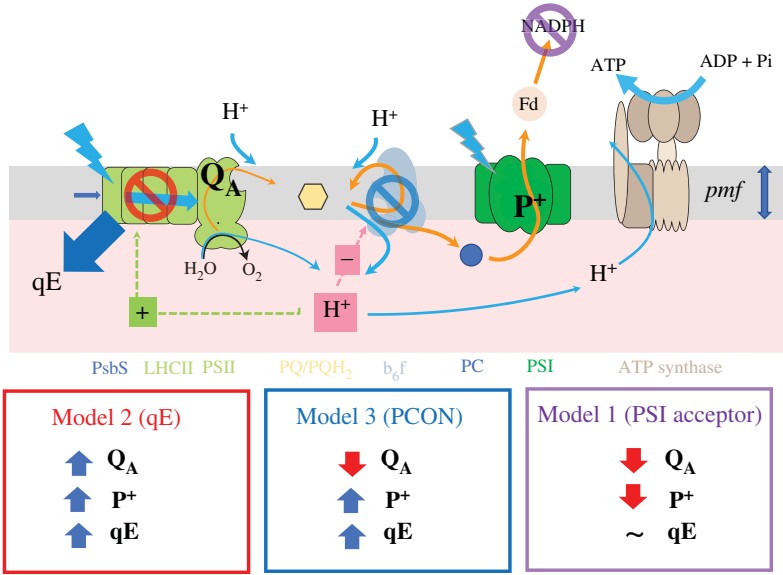

**Scheme 1.** Three basic mechanistic models describing proposed processes that can limit the LPs of photosynthetic and photoprotective mechanisms.

Scheme 1 illustrates three basic mechanistic models describing proposed processes that can limit the LPs of photosynthetic and photoprotective mechanisms. The models make qualitative predictions about how the actions of each mechanistic model will impact correlations between measured photosynthetic parameters, and thus can be used as a framework for interpreting the field data introduced in Results. The expected effects on the measured parameters are summarized in scheme 1, which shows specific effects of each model.

**Model 1: PSI acceptor-side limitations** (scheme 1, Model 1) where lack of $NADP^+$, ferredoxin or other PSI acceptors prevent further LEF. We expect this limitation to result in accumulation of electrons throughout the electron transfer chain, thus resulting in net reduction of $Q_A$ (decreasing qL) and $P_{700}^+$ (decreasing the 810 nm absorbance signal). The decreases in proton fluxes associated with back-up of electrons may, in addition, prohibit rapid, light-induced increases in *pmf*, lumen acidification and qE responses.

**Model 2: Increased NPQ** (scheme 1, Model 2) should decrease delivery of excitation energy to PSII (but not to PSI), resulting in net oxidation of $Q_A$ (increasing qL) and $P_{700}^+$ (increased 810 nm DIRK signal). Under some conditions, the NPQ will be rapidly induced by increased *pmf* and lumen acidification followed by activation of qE, which should be visible as increased $NPQ_{high-amb}$. Under other conditions, e.g. at higher $PAR_{amb}$, NPQ may already have been induced. If this NPQ is in the form of rapidly reversible qE, it should substantially decay during the 10 s dark recovery period, resulting in increased $NPQ_{high-rec}$. More slowly induced or relaxing forms of NPQ, including qI, qZ and long-lived qE, may be also present prior to and throughout the experiment. The forms should register as increases in $NPQ_{rec}$, but not in $NPQ_{high-amb}$ or $NPQ_{high-rec}$, but given that the high light and recovery periods were only 10 s long, our results do not allow us to distinguish among these possible forms.

**Model 3: Photosynthetic control** (PCON, scheme 1, Model 3). PCON results from the slowing of $PQH_2$ oxidation at the cytochrome $b_6f$ complex as the lumen becomes acidified. If PCON occurs without activation of qE, we expect a net reduction of $Q_A$ (decreasing qL) but a net oxidation of $P^+$ (increasing the 810 nm absorbance signal).

The qE and PCON models can be further subdivided [15,18]. In most cases, we expect lumen acidification accompanied by elevated *pmf*, reflected in an increased ECSt signal, which can be induced by increased proton influx into the lumen, due to increased LEF, increased CEF, or decreased conductivity of the thylakoid to protons ($g_H^+$) by slowing the ATP synthase, all of which can contribute to change in *pmf* under fluctuating light [11,18,54,55]. Alternatively, lumen acidification can also be associated with an increase in the fraction of *pmf* that is stored as ΔpH, by controlling the flow of counterions across the thylakoid membrane, altering the partitioning of *pmf* in ΔpH and Δψ [10,16,56]. In this case, acidification may occur with little or no increases in total *pmf*, or the rates of proton influx [57], though the current field-based data do not allow us to directly distinguish these possibilities.

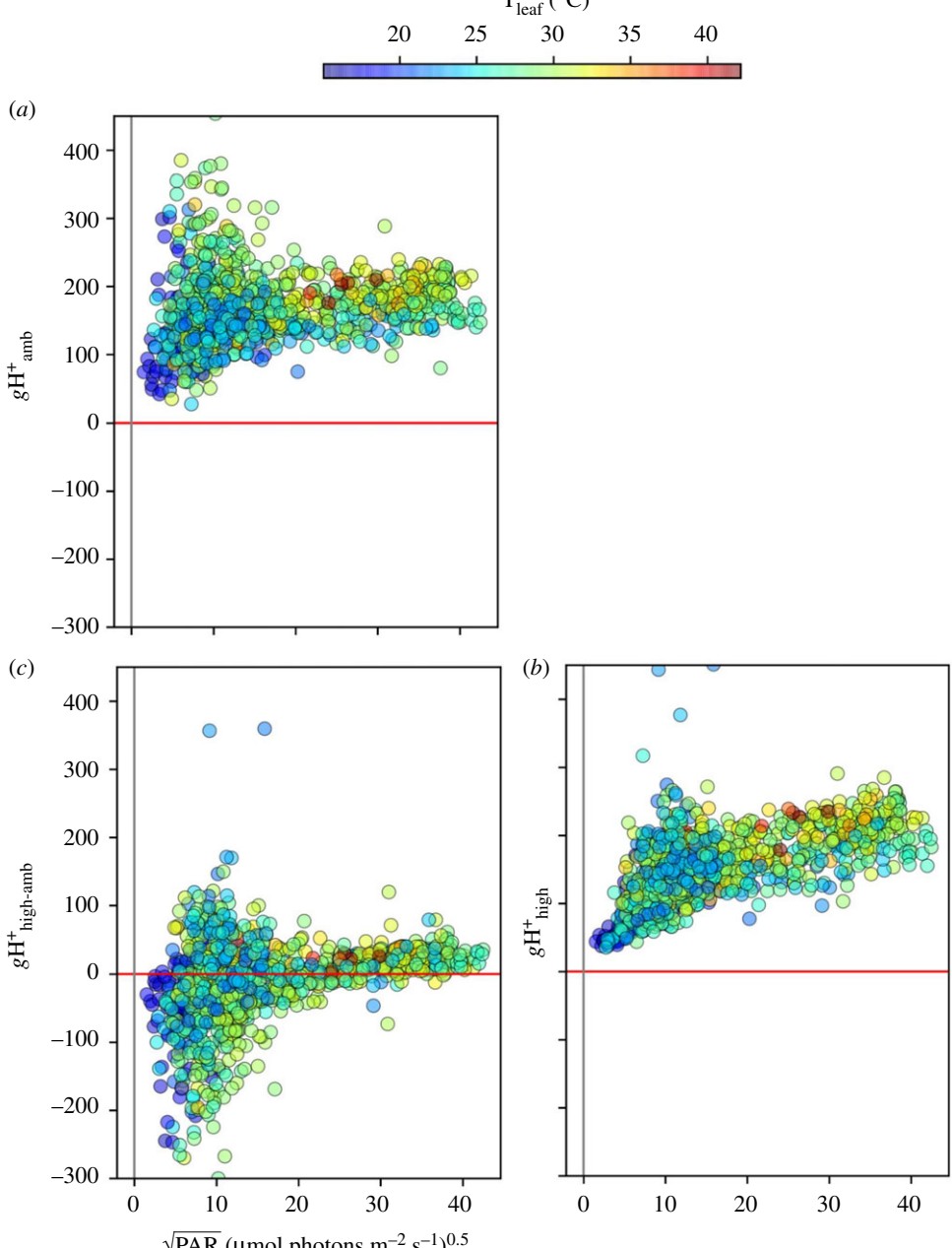

**Figure 7.** The relationship between light-induced thylakoid *pmf* and changes in $P_{700}$ redox state. Changes in the thylakoid *pmf* ($ECSt_{high-amb}$) were estimated using the ECSt parameter, and changes in $P_{700}^{+}$ were measured using the absorbance changes at 810 nm, as described in Material and methods, under ambient light ($ECSt_{amb}$, $P_{amb}^{+}$) and after 10 s of high light ($ECSt_{high}$, $P_{amb}^{+}$). The coloration of the points was set to a function of the square root of ambient PAR ($PAR_{amb}$).

These models, while not mutually exclusive, will tend to counteract each other, at least within a particular leaf. For instance, PSI acceptor-side limitations will tend to inhibit electron flow, thus decreasing proton flux and *pmf* generation. On the other hand, the generation of *pmf* will tend to slow electron flow (through Models 2 or 3), thus preventing the build-up of electrons on PSI electron acceptors. However, it is important to note that, in a survey-type experiment like ours, photosynthesis in different leaves may be limited by distinct processes, and thus any collection of samples may reflect various combinations of the above models.

## 4.4. Testing models for limitations in light potentials

By plotting MultispeQ parameters against each other, we can test for more detailed patterns of behaviours predicted by the above models. Figure 7 shows that $P_{high-amb}^{+}$ (high light-induced $P_{700}$

oxidation) was positively correlated with light-induced increases in *pmf* (ECSt$_{high-amb}$). Under all conditions, increasing PAR from PAR$_{amb}$ to PAR$_{high}$ resulted in a net oxidation of P$_{700}$, i.e. P$^+$$_{high-amb}$ was consistently positive. This behaviour is consistent with Models 2 (NPQ) or 3 (PCON), both of which predict a decrease in delivery of electrons from PSII to PSI. By contrast, we did not see evidence for high light-induced net *reduction* of P$_{700}$$^+$, i.e. values of negative P$^+$$_{high-amb}$, implying that Model 1 was not a major limitation to LEF LP. This does not exclude Model 1 from limiting photosynthesis in different species and conditions, as has been proposed to be important in chilling sensitive plants [58] as well as under pulse light [59] or in mutants that sufficiently acidify the lumen and activate PCON [21,23]. The apparent avoidance of Model 1 (or prevalence of Models 2 and 3) behaviour may reflect the 'tuning' of the light reactions to prevent the accumulation of reduced electron acceptors of PSI associated with photodamage [23], and the associated O$_2$ caused by build-up of electrons on PSI [60].

Overall, the behaviours seen in figure 7 are consistent with restrictions in electron flow to PSI imposed by increases in *pmf*, most likely through the acidification of the thylakoid lumen. In the case of Model 2 (rapid NPQ), this would be related to the induction of qE, while in Model 3 (PCON), this could be related to slowing of electron flow at the cytochrome b$_6$f complex.

Figure 8a further investigates this behaviour by plotting the dependence of high light-induced changes in P$_{700}$$^+$ (P$^+$$_{high-amb}$) with changes in Q$_A$ redox state (qL$_{high-amb}$). The expected theoretical changes in measurable parameters upon activation of the three models are indicated by the coloured boxes in the figure, and can be related to Models 1–3 in scheme 1:

— **Model 1** (violet box) predicts net **reduction** of P$_{700}$ (P$^+$$_{high-amb}$ < 0) and net **reduction** of Q$_A$ (qL$_{high-amb}$ < 0)
— **Model 2** (red box) predicts net **oxidation** of P$_{700}$ (P$^+$$_{high-amb}$ > 0) and net **oxidation** of Q$_A$ (qL$_{high-amb}$ > 0)
— **Model 3** (blue box) predicts net **oxidation** of P$_{700}$ (P$^+$$_{high-amb}$ > 0) but net **reduction** of Q$_A$ (qL$_{high-amb}$ < 0).

We observe behaviours consistent with both Models 2 and 3, suggesting that the behaviour of the system changed with conditions. Note that the boxes in figure 8a represent 'pure' behaviours, and it is possible that the effects of a particular mechanism may be intermediate, e.g. the responses may be limited by a combination of reduction of Q$_A$ and increased NPQ.

Figure 8b plots the dependence of NPQ$_{high-amb}$, which can be attributed to light-induced qE changes, on light-induced *pmf* changes (ECSt$_{high-amb}$). A generally positive correlation was observed between NPQ$_{high-amb}$ and ECSt$_{high-amb}$, but with high variability, especially at higher values. Applying the clustering obtained for figure 8a on top of the data in figure 8b, we see that this variability can be explained by the environmental conditions and the modes of behaviours. Specifically, we see clear evidence for condition-dependent suppression of rapid activation of qE in response to increases in *pmf*. Particularly, the sensitivities of NPQ$_{high-amb}$ to ECSt$_{high-amb}$, as indicated by the slopes in figure 8b, were smallest in clusters 1 (slope ∼ 1.6) and 2 (slope ∼ 17.7), which comprise those with Model 3-like behaviour and occurred at low T$_{leaf}$ and PAR$_{amb}$ values. Higher sensitivities of NPQ$_{high-amb}$ to ECSt$_{high-amb}$ were seen for clusters 3 (slope ∼ 28.1) and 4 (slope ∼ 35.1), which comprised those associated with Models 2 and intermediate, and occurred at higher T$_{leaf}$ and PAR$_{amb}$ values.

To assess what controlled the switch between Models 2 and 3, we performed GMM (using qL$_{high-amb}$, P$^+$$_{high-amb}$, T$_{leaf}$ as inputs). Four distinct clusters were observed (see symbol colours, figure 8a). Intercluster comparisons show that points in clusters 1 and 2 fell exclusively in the region predicted for Model 3. Cluster 3 fell entirely within the region predicted for Model 2. Cluster 4 extended between these regions, possibly indicating contributions from both mechanisms. The clusters falling in the Model 3 region were associated with relatively low T$_{leaf}$ (figure 8c) and PAR$_{amb}$ (figure 8d), compared with those associated with Model 2 or intermediate behaviours, suggesting that Model 2 prevailed at higher T$_{leaf}$ and/or PAR$_{amb}$, while Model 3 prevailed at lower values. Within the GMM clusters (electronic supplementary material, figure S13), qL$_{high-amb}$ was dependent predominantly on T$_{leaf}$ (cluster 3), PAR$_{amb}$ (cluster 1), or both (clusters 2 and 4). This dependence suggests that T$_{leaf}$ and PAR$_{amb}$ acted either independently or cooperatively, depending on conditions, affecting the propensity for photosynthesis to adopt Model 2 or 3 behaviours. As a first-order test of the robustness of these clusters by re-analysing randomly selected subpopulations of the data. As discussed in the legend to electronic supplementary material, figure S14, we obtained comparable results, i.e. that we would

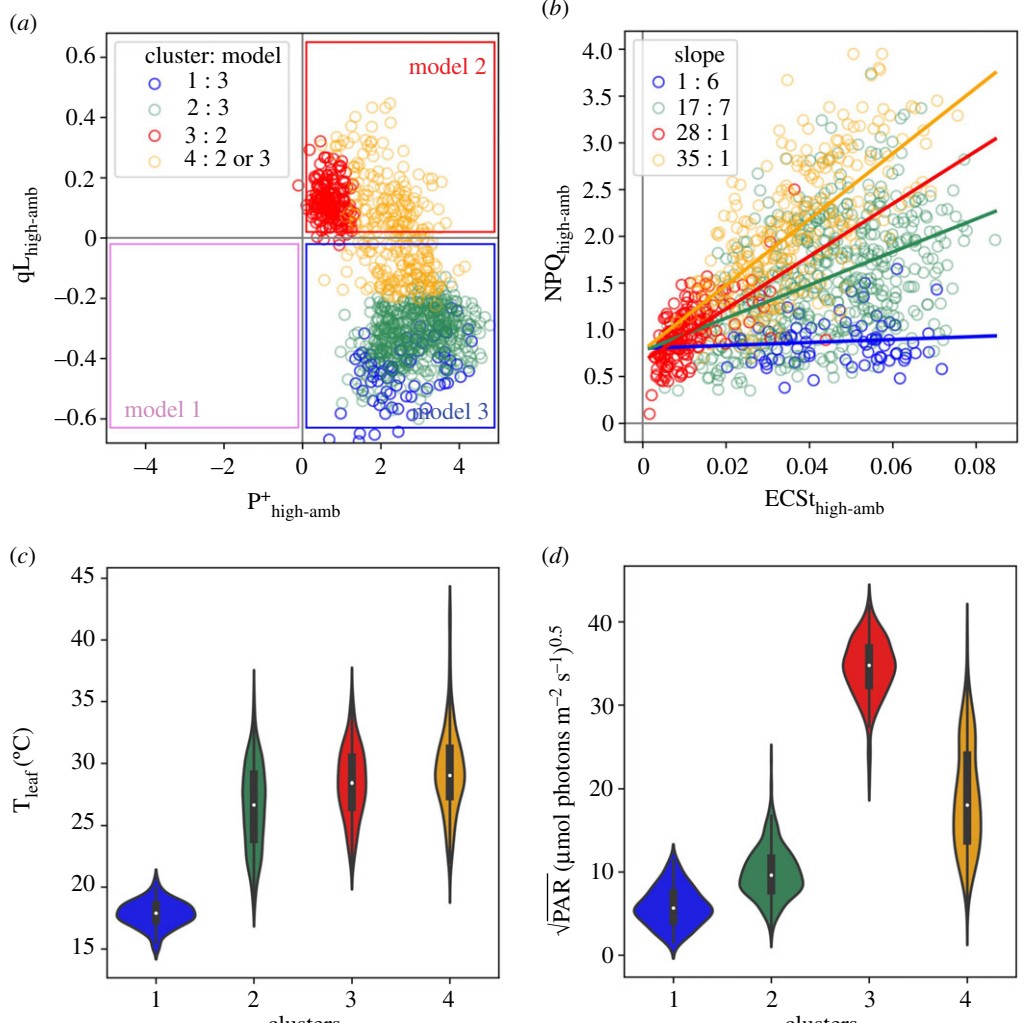

**Figure 8.** Relationships among measured parameters, predicted model behaviours and clustering. The relationships between light-induced changes in $Q_A$ redox state and P700 redox state (a), and between rapidly inducible NPQ and thylakoid *pmf* (b) and the leaf temperature (c) and PAR (d) dependencies of Gaussian mixture models (GMM) clusters. Changes in P700$^+$ (P$^+_{high-amb}$), $Q_A$ redox state (qL$_{high-amb}$), rapid changes in NPQ (NPQ$_{high-amb}$) and thylakoid *pmf* (ECSt$_{high-amb}$) were measured as described in Material and Methods. Data were clustered using the GMM approach described in the text, resulting in four distinct clusters, designated by the blue, green, red and ochre symbol colours (see legend in a). In (b), the slopes for the relationship between NPQ$_{high-amb}$ and ECSt$_{high-amb}$ were estimated by linear regression (slopes for clusters 1, 2, 3 and 4 were estimated to be 1.6, 17.7, 28.1 and 35.1, respectively). Panels (c,d) show distributions of (c) leaf temperatures (T$_{leaf}$) and (d) square root of ambient PAR for each cluster in (a,b).

interpret in similar ways, with as few subpopulations as small as 25% of the full dataset, suggesting that the clustering approach was reasonably robust.

The data in figure 8 show that, at lower T$_{leaf}$ and PAR$_{amb}$, qE activation was suppressed despite light-induced increases in *pmf*, and that this behaviour was associated with accumulation of electrons on $Q_A$ but oxidation of P$_{700}$ (figure 8a), suggesting that, under these conditions, light-induced increases in ΔpH caused slowing of the cytochrome $b_6f$ complex (PCON), but that the qE response lagged behind or was completely suppressed, leading to Model 3 behaviour. It is known that, initially after an abrupt increase in PAR, increased thylakoid *pmf* is stored as Δψ; conversion of Δψ to ΔpH is controlled by the movement of counterions across the thylakoid membrane, and protonation of lumenal proton buffering groups occurs over the seconds to tens of seconds time scale [25,61–63], and is dependent on the activities of various thylakoid ion transporters [9–11]. However, little is known about the natural diversity of Δψ/ΔpH balancing.

It has been shown that the lumen pH-dependencies of qE and PQH$_2$ oxidation by the cytochrome $b_6f$ complex are tightly coordinated, so that increased lumen acidity activates photoprotection prior to

PCON, presumably to prevent the accumulation of reduced $Q_A$ [15]. However, these experiments were performed under more slowly changing (near steady-state) conditions in the laboratory, and our results suggest that this coordination can be defeated under real-world conditions in the field, especially when $T_{leaf}$ is low and PAR fluctuates rapidly. This discoordination can have strong implications for photodamage, as it has been shown that high thylakoid *pmf* can greatly accelerate PSII recombination reactions, especially when $Q_A$ is reduced, leading to $^1O_2$ production [28,29,32]. It thus seems reasonable to suggest that the shift from qE to PCON at low $T_{leaf}$ will increase the rates of photodamage.

There are several possible mechanisms by which the response of qE can be uncoupled from increased *pmf*. Longer-term dependencies of NPQ on temperature have been reported under both field [64–66] and laboratory [67,68] conditions. The current work shows effects on rapid NPQ and LEF changes, which can be related to distinct mechanistic models. For example, it is known that the xanthophyll cycle is strongly temperature dependent, though the general observation is that zeaxanthin tends to accumulate at lower temperatures due to a slowing of the epoxidation of zeaxanthin [67–69]. Interestingly, we would expect the accumulation of zeaxanthin to augment, rather than suppress qE responses as we have observed in the current results. Lumen acidification may also be rate limiting for formation of qE. While rapid increase in light can result in nearly instantaneous increases in $\Delta\psi$, formation of $\Delta$pH and lumen acidification require counterion transport processes, which tend to be slow, and thus lumen acidification lags behind [25,29], and it is possible that this process is substantially slowed at low temperature. Other possible limitations include temperature-dependence of conformational rearrangement of antenna complexes following protonation of PsbS [70,71], which in turn may be related to the interactions among thylakoid proteins, lipids and ultrastructure [12,45,72,73]. The current data do not allow us to discriminate between these models, but the work suggests conditions and species under which such limitations occur, and how they may impact plant productivity or resilience.

# 5. Conclusion: current limitations and prospects for open science-led efforts to understand and improve photosynthesis

There are intense, ongoing efforts to improve photosynthesis, yet the importance of the responses of photosynthesis under fluctuating, real-world conditions are just now being recognized. In particular, we lack understanding of the extents and impacts of these responses, as well as their mechanisms and genomic control, which will be critical to achieving field-relevant improvements in efficiency and robustness, especially in a changing environment.

Here, we demonstrate methods and tools to assess the light responses of photosynthetic processes under real-world conditions, and use them to explore the factors that limit the capacity of plants to use or dissipate rapidly increased PAR. A major outcome is that, despite the complexities of field environments, clear behavioural patterns can be resolved, as long as the experiment contains a sufficient number of points taken over a large environmental space, and that includes environmental metadata. Such combinations of rapid measurements allowed us to test for various models over broad scales by looking for internally consistent relationships among the various measured parameters. For example, we observed no evidence for Model 1 (limitation at the acceptor side of PSI) behaviour in the current study, but we do not exclude the possibility in different species and/or different environmental conditions. The analysis supports the operation of Model 2, the rapid activation of NPQ resulting in net oxidation of $Q_A^-$ and Model 3, the strong activation of PCON, resulting in accumulation of $Q_A^-$. We surmised that Model 2 behaviour would be the most photoprotective, while Model 3 type behaviour would probably lead to photodamage, though we do not have independent endpoint measurements (e.g. yield, growth rates, etc.) to validate that the propensity for Model 3 behaviour has long-term consequences. Further, the models are not exclusive, and there will almost certainly be cases, e.g. cluster 4 in figure 8, where intermediate behaviours will be apparent, either because of co-limitations among multiple processes or heterogeneity between chloroplasts in the leaf samples.

We also emphasize that the data presented here were intended to introduce the approaches and methods, and thus leave a number of questions unanswered, but set up the approach to further study. The origins of these effects may include several classes of processes [31,74] that may differ under different conditions [75], including induction of downstream assimilatory reactions and metabolic pools [76,77], downstream sink reactions [78], redox regulation [79,80], balancing between the production and consumption of ATP and NADPH [1,49], ion homeostasis and regulation of thylakoid

*pmf* [25,81], low stomatal aperture that may lead to transient depletion of internal $CO_2$ levels. Distinguishing these will probably require more detailed phenotyping and biochemical [10,60,82] modelling [31] and genomics and genetics approaches [83].

The accessibility of the tools should allow larger numbers of researchers to answer these types of questions over a broader set of results. This approach was made possible by the combination of several open science advances. Collation of large amounts of data and metadata through the MultispeQ and PhotosynQ platforms [34], allowed us to explore the interdependencies of multiple responses and environmental conditions (metadata). The GMM methods allowed us to explore the interactions among multiple environmental parameters and photosynthetic responses, and test for the participation of distinct mechanistic models to explain the limitations to photosynthesis under field conditions, leading to the identification of distinct limitations in the rapid activation of NPQ and LEF at low temperature. Finally, making all tools, protocols and analytical methods available in directly usable forms, the project can be readily expanded to include multiple environments and species, as well as alternative models.

Data accessibility. Primary data are available on the photosynq.org site under the project 'rapid-ps-responses-pam-ecst-npqt-mint-dmk'. Data cleaning and analysis code is available in a GitHub repository (https://github.com/protonzilla/Light-Potentials-in-Field).

The data are provided in electronic supplementary material [84].

Authors' contributions. A.K. and D.M.K. designed the experiments. A.K. and H.T. conducted experiments. A.K., A.C., S.K. and D.M.K. analysed data. A.K., A.C., S.K., T.M. and D.M.K. contributed to the interpretations of data and writing the manuscript.

Competing interests. D.M.K. and S.K. are co-founders of PhotosynQ which maintains the PhotosynQ platforms and distributes and maintains the MultispeQ instruments. The current project was performed independently with no funding to or from the PhotosynQ organization.

Funding. Development of the protocols and data analysis methods was supported by the US Department of Energy (DOE), Office of Science, Basic Energy Sciences (BES) under Award no. DE-SC0007101 (The Energy Budget of Dynamic Photosynthesis) with additional support from Award no. DE-FG02-91ER20021 (Photosynthetic Energy Capture, Conversion and Storage: From Fundamental Mechanisms to Modular Engineering). The collection of field data by A.K. and H.T. was funded by the US National Science Foundation (1847193). D.M.K. received partial salary support from Michigan AgBioResearch.

Acknowledgements. The authors thank Dr Ute Armbruster, Thekla von Bismarck, Dr Nicholas Fisher, Dr Jennifer Johnson, Dr Thomas Avenson and Oliver Tessmer for valuable discussions and/or critical reading of the manuscript. We thank the reviewers for insightful comments and suggestions. We also thank numerous contributors to PhotosynQ datasets.

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
