## [Peer Review File · Royal Society Open Science]

Review History

RSOS-211102.R0 (Original submission)

Review form: Reviewer 1

Is the manuscript scientifically sound in its present form?

No

Are the interpretations and conclusions justified by the results?

Yes

Is the language acceptable?

Yes

Do you have any ethical concerns with this paper?

No

Have you any concerns about statistical analyses in this paper?

No

Recommendation?

Accept with minor revision (please list in comments)

Comments to the Author(s)

The authors obtained numerous photosynthetic data set with handy devices called MultispeQ V2.0. and analyzed these data using an unbiased statistical method. This reviewer enjoyed reading the manuscript.

Plant materials: The species name should be given. Given that there are many hybrids in this genus, it is necessary to give some more information. The location of MSU and the dates and season in which these measurements were conducted should be given. Addition of meteorological data during this period is also needed. For ordinary description of a plant stand, the height and leaf area index of the stand are also fundamental.

MutispeQ V2.0: As the authors are fully aware, the photon flux density per unit wavelength of far-red light is roughly comparable to that in orange to red light. Moreover, the FR/Red ratio dramatically increases with the depth from the canopy surface due to selective absorption of red light by green leaves. For examinations of P700 oxidation and PCON in the leaves in natural environment, the use of 655 nm LED only produces biased results. Not only PPFD level but also light quality may be considered for proper assessment of photosynthetic behaviours of the leaves in natural light. The authors dealt with the leaf angle applying the cosine law. However, when the light is incident on the leaf obliquely, reflectance increases, transmittance decreases, and the light path is lengthened. Thus, the situation is complicated. Also, this reviewer wonders the chamber volume and whether the air in the chamber was agitated. Stomatal responses may be neglected. However, if a leaf part is enclosed in a small volume of still air, CO₂ diffusion will be greatly suppressed.

Light potentials: The authors use this phrase in the title as well as in the abstract. Because this term is not used widely, a solid definition is needed.

Minor points

P. 19, 2nd para starting with A major outcome...: Is not it possible to analyze the present data with the meteorological data of the stand measured at the same time? For example, behaviours of two leaves under the same irradiance may be different depending on the environmental pre-histories of these leaves before the MultispeQ measurements.

P.19, 4 th para: Unbiased... is OK, but it may be worth comparing the top and bottom leaves (although subjective).

P. 25: Model 1 will be further unlikely if the authors take account of the presence of FR light in nature.

P.27: $\Delta \Psi$ is spontaneous while ΔpH is certainly slower. But, in high light it is a matter of a few second or so. More quantitative description is possible here.

The word 'data' is plural.

Review form: Reviewer 2

Is the manuscript scientifically sound in its present form?

Yes

Are the interpretations and conclusions justified by the results?

Yes

Is the language acceptable?

Yes

Do you have any ethical concerns with this paper?

Yes

Have you any concerns about statistical analyses in this paper?

No

Recommendation?

Major revision is needed (please make suggestions in comments)

Comments to the Author(s)

The paper entitled "Light Potentials of Photosynthetic Energy Storage in the Field: What limits the ability to use or dissipate rapidly increased light energy?" by Kanazawa and coworkers (RSOS-211102) describes an analysis a set of physiological parameters, testing different properties/responses of the photosynthetic apparatus, gathered over a large sample of plants in the field, hence subjected to environmental fluctuations.

In particular the authors make use of device allowing to monitor different parameters by optical methods, either employing fluorescence or absorption transient, which are substantially well-established proxies for the study of photosynthetic responses. Using this set of information gathered almost simultaneously on each sampled leaf, it is then possible to asses limitations arising, principally, from limitation of at the level of Photosystem I, Photosystem II or intermediate component in the electron transfer chain, as well a possible synergies/compensatory effects of these factors. Together with the physiological parameters the intensity of incident light ("PAR radiation") and the leaves temperature are measured as well, allowing thus to statistically correlated the physiological responses with the environmental status (at the time of the measurement).

To this end the author employ a non-supervised statistical algorithm capable of resolving correlations between the "environmental parameters, PAR intensity and leaves temperatures) for each of the physiological parameters estimated, and to distinguish, therefore, which of environmental condition is dominant and if an how, there is any inter-dependence (cross-correlation).

Based on the "grouping/sorting" of specific responses, it emerges that the factors which appears to limit photosynthesis in the field, i.e. for organism growth under non-controlled conditions, despite display a relatively large variability, as it may be expected, also show some distinctive features, which are particularly pronounced for low-PAR leaves (more often, but not always, corresponding also to low leaf temperatures). Under these conditions leaves in general appears to have a sub-optimal electron transport capacity, even when exposed to saturating intensity at the sampled area, and to have a strongly suppressed ability to develop regulative non-photochemical quenching at PSII level. Yet, in most conditions, for the organism tested (Mint) photosynthetic

electron transport appears to be limited by the plastoquinone shuttle oxido-reduction, hence mainly by PSII/Cytb6f turnover, whereas PSI is found to be substantially non-limiting.

The study is interesting in that it explores the possibility to determine photosynthetic-limiting factors under “ambient growth” conditions, henceforth eliminating possible bias brought about by specific “laboratory/greenhouse” growth conditions. Nonetheless there are a few points that would need clarification and which are listed in the below.

1- one general comment is that in the is not specified how large the sample (number of leaves/independent plants) shall be in order for the statistical approach employed to be robust. This is a key information if the method is indeed to be transferred in practice for phenotyping/lines/cultivar or even species selection in a given environment. One suggestion I could advance, would be that of “jack-knife” like approach, i.e. excluding an increasing percentage of data from the sample (randomly sorted), and the very whether the clustering converges or not, within reasonable margin of confidence.

2- A point which appears to have been overlooked is the influence of “incident light spectrum” on top of the PAR intensity per se. especially for plants which tend to form rather dense canopies, and Mint appears to do so, low intensity might be not only the result of clouding or other “sky” shading factors, rather because of leave-mediated filtering. Yet, in this case, the colour of the incident would also be much different than that of direct exposure, such as the canopy top. I would concede it may not be possible to check this parameter at posteriori. Nonetheless, the possibility that the PAR spectrum might play a role in the observed behaviour, shall at least be mentioned, since the largest limitations appears to have been determine at low-PAR when filtering is predictably more pronounced too. This might as well be correlated to leaf temperature too.

3- I would not agree that the fast reversible qE component is always the dominant parameters in the data observed. Although it is certainly true that the authors observe a large rapidly reversible quenching fraction (that is assignable to qE), for large induction of NPQ (values reacting 3-4), and upon a relaxation of 2 NPQ units, it means the leaves are still significantly quenched to NPQ equivalents of 1-to-2. This is a 50%-75% excited state quenching, hence a large quenching. In terms of excited state the relaxed part, is, actually, much less. To better evaluate the effect of excited state quenching (rather than the quencher concentration, to which the “NPQ” parameter is related), it might far better to use the “yield of quenching” parameters ($=1-F_m'/F_m$), this is constrained to range from 0-to-1 (being a yield).

Some specific points:

1- summary, line 35 “efficient light capture”, I trust this shall be “light conversion”.

2- introduction line 33-34 and in other places: “photo-protective capacity”. Whereas I fully agree with the statement concerning balancing of non-photochemical and photochemical quenching, the extent by which NPQ effectively protects PSII is yet to be unambiguously determined, with suggestion it could even be rather modest, unless quenching is very large. It would certainly act as a regulative process, in terms of electron transport, irrespective of the direct protective effect on light-induced damage.

3- page 10-11, often the terms “phenotype” appears, which, although understandable as a “functional” term, is a bit odd in this contest, as it might refer even to different leaves within the same plant, and hence clash with the most common use of the term. I might suggest the term “response behaviours” or something alike.

4- Figure 3, 4 and in general. It might be interesting to present, even in the supplementary information a direct correlation plot of the measured parameters, i.e. LEF vs NPQ, LEF vs ECS, etc

5- Page 19, line 19 “rapidly variable”, I trust this shall be “reversible”

6- Page 22, paragraph from line 14. The statement seems broadly unjustified. It is clear, even from clustering, that there is a correlation between the environmental factors and the measured parameters. In case of controlled environment the response measured shall still be correct, but refer preferentially, to the specific growth conditions employed. Furthermore later on the authors themselves stated that further investigation, possibly by controlling the conditions (presumably to limit the number of stochastic "environmental" parameter) shall be needed to better understand the trends observed in this paper. It would be fairer to say that the comparison of "environmental fluctuations" (not all of which could be monitor) and, possibly a set of "controlled conditions" (rather than just "a" controlled condition), would be more informative in the future.

7- same page, line 37, as in the general comments. The majority of quencher appears to be of the qE type, the majority of the quenching, as decrease in excited state population, not necessarily so, depending on the individual sample. Residual, slowly reversible/irreversible quenching, delta-pH unrelated, appears to be dominant, in terms of dissipation in a good fraction of the sampled leaves.

8- Page 26 line 14, and following. "net reduction/net oxidation", etc. This is with respect to non-regulative process taking places.

9- page 20. Same as point 6

10- Conclusion. I'd recommend to specify, what Model 1/2/3 are here, i.e. (control by qE, by photosynthetic control, PSI donor/acceptor limitation.

11- page 28: "upcoming paper", sound somewhat mysterious for an "open science" journal. Details on how the presented data and methods are effectively intended to be made broadly accessible shall be specified.

12- the authors often refer to the protocol employed with "quasi-commercial" acronyms. Yet, since the protocols and approaches rely on the quantities measured, which, provided estimated correctly, can be assessed by different commercial or laboratory-developed instruments, there's no need or reason, to refer to "specific" commercial/jargon name for them.

13- I find it a shame that most of the parameter clustering is presented in the supplementary material only, as it is informative and successive analysis rely on it. Nonetheless is accessible. Yet, I have a slight concern. Whereas some of cross-correlation are very clear, some other, albeit appears to be significant (according exactly to which significance test, since it is not specified?), might be in part biased by a few outliers (e.g cluster 6 figure S3 bottom panels of the matrix, S4, bottom corner, and so on).

Review form: Reviewer 3 (Marian Brestic)

Is the manuscript scientifically sound in its present form?

Yes

Are the interpretations and conclusions justified by the results?

Yes

Is the language acceptable?

Yes

Do you have any ethical concerns with this paper?

Yes

Have you any concerns about statistical analyses in this paper?

No

Recommendation?

Accept as is

Comments to the Author(s)

The title and subject of the manuscript are very interesting from a methodological and practical point of view, suitable and adequate. The scientific content contributes to the space in which it develops. The work was provided with a sufficient level of scientific novelty.

This seems to be well-conducted research and to have clear, repeatable results. The authors studied the responses of plant photosynthesis to rapid fluctuations in environmental conditions. Authors suggest With decreasing leaf temperature or PAR, limitations to photosynthesis during high light fluctuations shifted from rapidly-induced NPQ to photosynthetic control (PCON) of electron flow at the cytochrome b6 f complex. At low temperatures, high light induced lumen acidification, but did not induce NPQ, leading to accumulation of reduced electron transfer intermediates, a situation likely to induce photodamage, and represents a potential target for improving the efficiency and robustness of photosynthesis.

I have no critical comments. Paper is very innovative. The structure of the paper is logical and the results are well reproduced. The introduction and discussion are well organized. Results reported have not been published elsewhere. Conclusions are presented in an appropriate fashion and are supported by the data. I think the overall concept is interesting and potentially important. I suggest reading/include some references of W. Huang, Ch. Miyake, Takagi, L. Ferroni.

I recommend to ACCEPT the paper for publication.

Decision letter (RSOS-211102.R0)

Dear Dr Kramer

The Editors assigned to your paper RSOS-211102 "Light Potentials of Photosynthetic Energy Storage in the Field: What limits the ability to use or dissipate rapidly increased light energy?" have now received comments from reviewers and would like you to revise the paper in accordance with the reviewer comments and any comments from the Editors. Please note this decision does not guarantee eventual acceptance.

Please submit your revised manuscript and required files (see below) no later than 21 days from today's (ie 13-Sep-2021) date. Note: the ScholarOne system will 'lock' if submission of the revision is attempted 21 or more days after the deadline. If you do not think you will be able to meet this deadline please contact the editorial office immediately.

on behalf of Prof Malcolm White (Subject Editor)
openscience@royalsociety.org

Associate Editor Comments to Author:

Thank you for the interesting paper - as you'll see, we've three comments received, and the reviewers find much to like in your work, but, as it stands, there are enough matters that need to be addressed that we would ask you to revise the manuscript to carefully take into account the comments made.

Reviewer comments to Author:

Reviewer: 1

Comments to the Author(s)

The authors obtained numerous photosynthetic data set with handy devices called MultispeQ V2.0. and analyzed these data using an unbiased statistical method. This reviewer enjoyed reading the manuscript.

Plant materials: The species name should be given. Given that there are many hybrids in this genus, it is necessary to give some more information. The location of MSU and the dates and season in which these measurements were conducted should be given. Addition of meteorological data during this period is also needed. For ordinary description of a plant stand, the height and leaf area index of the stand are also fundamental.

MutispeQ V2.0: As the authors are fully aware, the photon flux density per unit wavelength of far-red light is roughly comparable to that in orange to red light. Moreover, the FR/Red ratio dramatically increases with the depth from the canopy surface due to selective absorption of red light by green leaves. For examinations of P700 oxidation and PCON in the leaves in natural environment, the use of 655 nm LED only produces biased results. Not only PPFD level but also light quality may be considered for proper assessment of photosynthetic behaviours of the leaves in natural light. The authors dealt with the leaf angle applying the cosine law. However, when the light is incident on the leaf obliquely, reflectance increases, transmittance decreases, and the light path is lengthened. Thus, the situation is complicated. Also, this reviewer wonders the

chamber volume and whether the air in the chamber was agitated. Stomatal responses may be neglected. However, if a leaf part is enclosed in a small volume of still air, CO₂ diffusion will be greatly suppressed.

Light potentials: The authors use this phrase in the title as well as in the abstract. Because this term is not used widely, a solid definition is needed.

Minor points

P. 19, 2nd para starting with A major outcome...: Is not it possible to analyze the present data with the meteorological data of the stand measured at the same time? For example, behaviours of two leaves under the same irradiance may be different depending on the environmental pre-histories of these leaves before the MultispeQ measurements.

P.19, 4 th para: Unbiased... is OK, but it may be worth comparing the top and bottom leaves (although subjective).

P. 25: Model 1 will be further unlikely if the authors take account of the presence of FR light in nature.

P.27: $\Delta \Psi$ is spontaneous while ΔpH is certainly slower. But, in high light it is a matter of a few second or so. More quantitative description is possible here.

The word 'data' is plural.

Reviewer: 2

Comments to the Author(s)

The paper entitled "Light Potentials of Photosynthetic Energy Storage in the Field: What limits the ability to use or dissipate rapidly increased light energy?" by Kanazawa and coworkers (RSOS-211102) describes an analysis a set of physiological parameters, testing different properties/responses of the photosynthetic apparatus, gathered over a large sample of plants in the field, hence subjected to environmental fluctuations.

In particular the authors make use of device allowing to monitor different parameters by optical methods, either employing fluorescence or absorption transient, which are substantially well-established proxies for the study of photosynthetic responses. Using this set of information gathered almost simultaneously on each sampled leaf, it is then possible to asses limitations arising, principally, from limitation of at the level of Photosystem I, Photosystem II or intermediate component in the electron transfer chain, as well a possible synergies/compensatory effects of these factors. Together with the physiological parameters the intensity of incident light ("PAR radiation") and the leaves temperature are measured as well, allowing thus to statistically correlated the physiological responses with the environmental status (at the time of the measurement).

To this end the author employ a non-supervised statistical algorithm capable of resolving correlations between the "environmental parameters, PAR intensity and leaves temperatures) for each of the physiological parameters estimated, and to distinguish, therefore, which of environmental condition is dominant and if an how, there is any inter-dependence (cross-correlation).

Based on the "grouping/sorting" of specific responses, it emerges that the factors which appears to limit photosynthesis in the field, i.e. for organism growth under non-controlled conditions, despite display a relatively large variability, as it may be expected, also show some distinctive

features, which are particularly pronounced for low-PAR leaves (more often, but not always, corresponding also to low leaf temperatures). Under these conditions leaves in general appears to have a sub-optimal electron transport capacity, even when exposed to saturating intensity at the sampled area, and to have a strongly suppressed ability to develop regulative non-photochemical quenching at PSII level. Yet, in most conditions, for the organism tested (Mint) photosynthetic electron transport appears to be limited by the plastoquinone shuttle oxido-reduction, hence mainly by PSII/Cytb6f turnover, whereas PSI is found to be substantially non-limiting.

The study is interesting in that it explores the possibility to determine photosynthetic-limiting factors under “ambient growth” conditions, henceforth eliminating possible bias brought about by specific “laboratory/greenhouse” growth conditions. Nonetheless there are a few points that would need clarification and which are listed in the below.

1- one general comment is that in the is not specified how large the sample (number of leaves/independent plants) shall be in order for the statistical approach employed to be robust. This is a key information if the method is indeed to be transferred in practice for phenotyping/lines/cultivar or even species selection in a given environment. One suggestion I could advance, would be that of “jack-knife” like approach, i.e. excluding an increasing percentage of data from the sample (randomly sorted), and the very whether the clustering converges or not, within reasonable margin of confidence.

2- A point which appears to have been overlooked is the influence of “incident light spectrum” on top of the PAR intensity per se. especially for plants which tend to form rather dense canopies, and Mint appears to do so, low intensity might be not only the result of clouding or other “sky” shading factors, rather because of leave-mediated filtering. Yet, in this case, the colour of the incident would also be much different than that of direct exposure, such as the canopy top. I would concede it may not be possible to check this parameter at posteriori. Nonetheless, the possibility that the PAR spectrum might play a role in the observed behaviour, shall at least be mentioned, since the largest limitations appears to have been determine at low-PAR when filtering is predictably more pronounced too. This might as well be correlated to leaf temperature too.

3- I would not agree that the fast reversible qE component is always the dominant parameters in the data observed. Although it is certainly true that the authors observe a large rapidly reversible quenching fraction (that is assignable to qE), for large induction of NPQ (values reacting 3-4), and upon a relaxation of 2 NPQ units, it means the leaves are still significantly quenched to NPQ equivalents of 1-to-2. This is a 50%-75% excited state quenching, hence a large quenching. In terms of excited state the relaxed part, is, actually, much less. To better evaluate the effect of excited state quenching (rather than the quencher concentration, to which the “NPQ” parameter is related), it might far better to use the “yield of quenching” parameters ($=1-Fm'/Fm$), this is constrained to range from 0-to-1 (being a yield).

Some specific points:

1- summary, line 35 “efficient light capture”, I trust this shall be “light conversion”.

2- introduction line 33-34 and in other places: “photo-protective capacity”. Whereas I fully agree with the statement concerning balancing of non-photochemical and photochemical quenching, the extent by which NPQ effectively protects PSII is yet to be unambiguously determined, with suggestion it could even be rather modest, unless quenching is very large. It would certainly act as a regulative process, in terms of electron transport, irrespective of the direct protective effect on light-induced damage.

3- page 10-11, often the terms “phenotype” appears, which, although understandable as a “functional” term, is a bit odd in this contest, as it might refer even to different leaves within the same plant, and hence clash with the most common use of the term. I might suggest the term “response behaviours” or something alike.

4- Figure 3, 4 and in general. It might be interesting to present, even in the supplementary information a direct correlation plot of the measured parameters, i.e. LEF vs NPQ, LEF vs ECS, etc

5- Page 19, line 19 “rapidly variable”, I trust this shall be “reversible”

6- Page 22, paragraph from line 14. The statement seems broadly unjustified. It is clear, even from clustering, that there is a correlation between the environmental factors and the measured parameters. In case of controlled environment the response measured shall still be correct, but refer preferentially, to the specific growth conditions employed. Furthermore later on the authors themselves stated that further investigation, possibly by controlling the conditions (presumably to limit the number of stochastic “environmental” parameter) shall be needed to better understand the trends observed in this paper. It would be fairer to say that the comparison of “environmental fluctuations” (not all of which could be monitor) and, possibly a set of “controlled conditions” (rather than just “a” controlled condition), would be more informative in the future.

7- same page, line 37, as in the general comments. The majority of quencher appears to be of the qE type, the majority of the quenching, as decrease in excited state population, not necessarily so, depending on the individual sample. Residual, slowly reversible/irreversible quenching, delta-pH unrelated, appears to be dominant, in terms of dissipation in a good fraction of the sampled leaves.

8- Page 26 line 14, and following. “net reduction/net oxidation”, etc. This is with respect to non-regulative process taking places.

9- page 20. Same as point 6

10- Conclusion. I’d recommend to specify, what Model 1/2/3 are here, i.e. (control by qE, by photosynthetic control, PSI donor/acceptor limitation).

11- page 28: “upcoming paper”, sound somewhat mysterious for an “open science” journal. Details on how the presented data and methods are effectively intended to be made broadly accessible shall be specified.

12- the authors often refer to the protocol employed with “quasi-commercial” acronyms. Yet, since the protocols and approaches rely on the quantities measured, which, provided estimated correctly, can be assessed by different commercial or laboratory-developed instruments, there's no need or reason, to refer to "specific" commercial/jargon name for them.

13- I find it a shame that most of the parameter clustering is presented in the supplementary material only, as it is informative and successive analysis rely on it. Nonetheless is accessible. Yet, I have a slight concern. Whereas some of cross-correlation are very clear, some other, albeit appears to be significant (according exactly to which significance test, since it is not specified?), might be in part biased by a few outliers (e.g cluster 6 figure S3 bottom panels of the matrix, S4, bottom corner, and so on).

Reviewer: 3

Comments to the Author(s)

The title and subject of the manuscript are very interesting from a methodological and practical point of view, suitable and adequate. The scientific content contributes to the space in which it develops. The work was provided with a sufficient level of scientific novelty.

This seems to be well-conducted research and to have clear, repeatable results. The authors studied the responses of plant photosynthesis to rapid fluctuations in environmental conditions. Authors suggest With decreasing leaf temperature or PAR, limitations to photosynthesis during high light fluctuations shifted from rapidly-induced NPQ to photosynthetic control (PCON) of electron flow at the cytochrome b6 f complex. At low temperatures, high light induced lumen acidification, but did not induce NPQ, leading to accumulation of reduced electron transfer

intermediates, a situation likely to induce photodamage, and represents a potential target for improving the efficiency and robustness of photosynthesis.

I have no critical comments. Paper is very innovative. The structure of the paper is logical and the results are well reproduced. The introduction and discussion are well organized. Results reported have not been published elsewhere. Conclusions are presented in an appropriate fashion and are supported by the data. I think the overall concept is interesting and potentially important. I suggest reading/include some references of W. Huang, Ch. Miyake, Takagi, L. Ferroni.

I recommend to ACCEPT the paper for publication.

===PREPARING YOUR MANUSCRIPT===

===PREPARING YOUR REVISION IN SCHOLARONE===

Please ensure that you include a summary of your paper at Step 2 'Type, Title, & Abstract'. This should be no more than 100 words to explain to a non-scientific audience the key findings of your

research. This will be included in a weekly highlights email circulated by the Royal Society press office to national UK, international, and scientific news outlets to promote your work.

Author's Response to Decision Letter for (RSOS-211102.R0)

See Appendix A.

RSOS-211102.R1 (Revision)

Review form: Reviewer 1

Is the manuscript scientifically sound in its present form?

Yes

Are the interpretations and conclusions justified by the results?

Yes

Is the language acceptable?

Yes

Do you have any ethical concerns with this paper?

No

Have you any concerns about statistical analyses in this paper?

No

Recommendation?

Accept as is

Comments to the Author(s)

The revised version reads well.

Review form: Reviewer 2

Is the manuscript scientifically sound in its present form?

Yes

Are the interpretations and conclusions justified by the results?

Yes

Is the language acceptable?

Yes

Do you have any ethical concerns with this paper?

No

Have you any concerns about statistical analyses in this paper?

No

Recommendation?

Accept with minor revision (please list in comments)

Comments to the Author(s)

Either in the revised version of the paper entitled "Light Potentials of Photosynthetic Energy Storage in the Field: What limits the ability to use or dissipate rapidly increased light energy?"

(RSOS-211102.R1) or through the point-by-point answer to reviewer comments, the author have addressed carefully and thoughtfully the majority of the point raised.

Although I would therefore consider the paper recommendable for publication, I would also like to raise two minor points, already signalled in the previous comments, that I trust can be addressed pretty straightforwardly by the authors.

Original point 12- In the answer the author states "...there are so many variations of measurements in the field and re-describing when used complicates the text. Such nomenclatures are common in fields such as EPR (e.g., ESEEM) or mass spectroscopy (e.g., MALDI TOF)."

I trust the author had no intention of making "commercials" for the instrument/methods. Yet, what they are discussing is a combined analysis from different experimental approaches and does not related to the acronyms used as counter-example which are proper spectroscopic methods (i.e. ESEEM, ENDOR, DEER, etc.. in EPR are specific MW/RF pulse sequences to investigate electron-nuclear or electron-electron interactions; same as in mass spectroscopy). What's discussed/proposed here is more of combination of methods, not simultaneously, but sequentially applied to the material, and that can be (and likely will) further expended, instrumental flexibility allowing.

- Original main point 3 - I trust the authors have only partially addressed the point raised. As correctly state, there are two issues, one concerns the relation between quenching and inhibition: that has properly addressed within the paper framework. The second is the impact of fraction of reversible/non (rapidly) reversible NPQ on regulation in broad sense. The point raised was that is not so much the fraction of reversible NPQ, rather the extent of residual quenching (after a dark/low light) transition, that likely represent the key parameters. To give two limit scenarios, one can thing a NPQ which develops to a level of 2, and fully recovers. The other in which develops to a level of 4 and recovers to a level of 2. In the first case the systems, after recovery, is fully unquenched and photochemical yield is therefore unaffected (this will be the case for a any "small" value of NPQ, since the rough equivalent for photochemical quenching is around 4-5). In the other case of large quenching development (for the same recovery), the system after the transition remains with a non-photochemical quencher which, to some extent, is in principle capable of competing (kinetically) with photochemistry instead.

I on the other hand fully disagree with the statement that the $\phi(\text{NPQ})$ parameter depends on photochemistry (or that it does differently from the NPQ one). If, as to my knowledge, NPQ is defined as $\text{NPQ} = F_m / F_m' = 1$ and $\phi(\text{NPQ}) = 1 - F_m' / F_m$ they contain exactly the same terms, all estimated at "closed centres" where photochemistry shall not be a factor. If it were to be, it would be for both parameters. It is just NPQ is proportional to the quencher(s) concentration(s) and $\phi(\text{NPQ})$ is a quenching "yield-like" parameters, varying from 0-1.

Decision letter (RSOS-211102.R1)

Dear Dr Kramer

On behalf of the Editors, we are pleased to inform you that your Manuscript RSOS-211102.R1 "Light Potentials of Photosynthetic Energy Storage in the Field: What limits the ability to use or

dissipate rapidly increased light energy?" has been accepted for publication in Royal Society Open Science subject to minor revision in accordance with the referees' reports. Please find the referees' comments along with any feedback from the Editors below my signature.

Please submit your revised manuscript and required files (see below) no later than 7 days from today's (ie 05-Nov-2021) date. Note: the ScholarOne system will 'lock' if submission of the revision is attempted 7 or more days after the deadline. If you do not think you will be able to meet this deadline please contact the editorial office immediately.

on behalf of Malcolm White (Subject Editor)
openscience@royalsociety.org

Associate Editor Comments to Author:

Comments to the Author:

The paper appears to be largely ready for publication, but the second reviewer has a couple of what appear to be relatively minor questions/comments that should be addressed by the authors. Please ensure that you respond to these comments and amend the manuscript accordingly.

Reviewer comments to Author:

Reviewer: 1

Comments to the Author(s)

The revised version reads well.

Reviewer: 2

Comments to the Author(s)

Either in the revised version of the paper entitled "Light Potentials of Photosynthetic Energy Storage in the Field: What limits the ability to use or dissipate rapidly increased light energy?" (RSOS-211102.R1) or through the point-by-point answer to reviewer comments, the author have addressed carefully and thoughtfully the majority of the point raised.

Although I would therefore consider the paper recommendable for publication, I would also like to raise two minor points, already signalled in the previous comments, that I trust can be addressed pretty straightforwardly by the authors.

Original point 12- In the answer the author states "...there are so many variations of measurements in the field and re-describing when used complicates the text. Such nomenclatures are common in fields such as EPR (e.g., ESEEM) or mass spectroscopy (e.g., MALDI TOF)."

I trust the author had no intention of making "commercials" for the instrument/methods. Yet, what they are discussing is a combined analysis from different experimental approaches and does not related to the acronyms used as counter-example which are proper spectroscopic methods (i.e. ESEEM, ENDOR, DEER, etc.. in EPR are specific MW/RF pulse sequences to investigate electron-nuclear or electron-electron interactions; same as in mass spectroscopy). What's discussed/proposed here is more of combination of methods, not simultaneously, but sequentially applied to the material, and that can be (and likely will) further expended, instrumental flexibility allowing.

- Original main point 3 - I trust the authors have only partially addressed the point raised. As correctly state, there are two issues, one concerns the relation between quenching and inhibition: that has properly addressed within the paper framework. The second is the impact of fraction of reversible/non (rapidly) reversible NPQ on regulation in broad sense. The point raised was that is not so much the fraction of reversible NPQ, rather the extent of residual quenching (after a dark/low light) transition, that likely represent the key parameters. To give two limit scenarios, one can thing a NPQ which develops to a level of 2, and fully recovers. The other in which develops to a level of 4 and recovers to a level of 2. In the first case the systems, after recovery, is fully unquenched and photochemical yield is therefore unaffected (this will be the case for a any "small" value of NPQ, since the rough equivalent for photochemical quenching is around 4-5). In the other case of large quenching development (for the same recovery), the system after the transition remains with a non-photochemical quencher which, to some extent, is in principle capable of competing (kinetically) with photochemistry instead.

I on the other hand fully disagree with the statement that the $\phi(\text{NPQ})$ parameter depends on photochemistry (or that it does differently from the NPQ one). If, as to my knowledge, NPQ is defined as $\text{NPQ} = F_m / F_m' = 1$ and $\phi(\text{NPQ}) = 1 - F_m' / F_m$ they contain exactly the same terms, all estimated at "closed centres" where photochemistry shall not be a factor. If it were to be, it would be for both parameters. It is just NPQ is proportional to the quencher(s) concentration(s) and $\phi(\text{NPQ})$ is a quenching "yield-like" parameters, varying from 0-1.

===PREPARING YOUR MANUSCRIPT===

one version should clearly identify all the changes that have been made (for instance, in coloured highlight, in bold text, or tracked changes);

===PREPARING YOUR REVISION IN SCHOLARONE===

-- Ensure that your data access statement meets the requirements at <https://royalsociety.org/journals/authors/author-guidelines/#data>. You should ensure that you cite the dataset in your reference list. If you have deposited data etc in the Dryad repository, please only include the 'For publication' link at this stage. You should remove the 'For review' link.

-- If you are requesting an article processing charge waiver, you must select the relevant waiver option (if requesting a discretionary waiver, the form should have been uploaded, see 'File upload' above).

-- If you have uploaded any electronic supplementary (ESM) files, please ensure you follow the guidance at <https://royalsociety.org/journals/authors/author-guidelines/#supplementary-material> to include a suitable title and informative caption. An example of appropriate titling and captioning may be found at https://figshare.com/articles/Table_S2_from_Is_there_a_trade-off_between_peak_performance_and_performance_breadth_across_temperatures_for_aerobic_scope_in_teleost_fishes_/3843624.

Author's Response to Decision Letter for (RSOS-211102.R1)

See Appendix B

Decision letter (RSOS-211102.R2)

Dear Dr Kramer,

I am pleased to inform you that your manuscript entitled "Light Potentials of Photosynthetic Energy Storage in the Field: What limits the ability to use or dissipate rapidly increased light energy?" is now accepted for publication in Royal Society Open Science.

Please remember to make any data sets or code libraries 'live' prior to publication, and update any links as needed when you receive a proof to check - for instance, from a private 'for review'

URL to a publicly accessible 'for publication' URL. It is good practice to also add data sets, code and other digital materials to your reference list.

on behalf of Prof Malcolm White (Subject Editor)
openscience@royalsociety.org

Appendix A

Dear Editor,

We want to thank the Editor and each of the reviewers for their insightful and positive comments. We are very gratified that they took the time to very thoughtfully read and digest the work, especially given the breadth of the subject matter. We have added an acknowledgement to their contributions.

We have responded to each comment as detailed below and revised the manuscript accordingly, including additional experimentation, statistical analyses and more detailed discussion.

We hope that the revised manuscript will be acceptable for publication.

Please note that, in the following, our replies to the comments are indicated in RED. Changes made to the text in response to these comments are highlighted in YELLOW in the MARKED COPY of the revised manuscript. The marked copy also has changes needed for resubmission, e.g., the figures and legends are moved from the main text.

Sincerely yours,

David M. Kramer

Responses to Comments

Reviewer: 1

Comments to the Author(s)

1) The authors obtained numerous photosynthetic data set with handy devices called MultispeQ V2.0. and analyzed these data using an unbiased statistical method. This reviewer enjoyed reading the manuscript.

Reply: We thank the reviewer for this comment! This is especially gratifying given the broad range of disciplines that are covered in the paper.

2) Plant materials: The species name should be given. Given that there are many hybrids in this genus, it is necessary to give some more information.

Reply: Plants were common garden spearmint, *Mentha spicata*. This description was added.

3) The location of MSU and the dates and season in which these measurements were conducted should be given.

Reply: The location was described in the manuscript and GPS coordinates for each measurement were included in the online data set. To clarify, we have added the following to the text:

In the manuscript p.4--MSU Horticulture Gardens (East Lansing, MI, USA); in the manuscript p.5--The experiment took place over a nine-day experimental window between July 21 and August 2, 2019 (Figure S1A), sampling a range of times of day (between 5:30 AM and 6:40 PM local time).

Addition of meteorological data during this period is also needed.

Reply: For the weather data, the link to the MSU Enviroweather web site (<https://mawn.geo.msu.edu/station.asp?id=msu>, MSU Horticulture Teaching & Research Center, Latitude: 42.6734, Longitude: -84.4870, Elevation: 264m) is added.

4) For ordinary description of a plant stand, the height and leaf area index of the stand are also fundamental.

Reply: The following sentences were added. "Plants were up to 1 m in height. The leaf area index was not measured in this experiment."

5) MutispeQ V2.0: As the authors are fully aware, the photon flux density per unit wavelength of far-red light is roughly comparable to that in orange to red light. Moreover, the FR/Red ratio dramatically increases with the depth from the canopy surface due to selective absorption of red light by green leaves. For examinations of P700 oxidation and PCON in the leaves in natural environment, the use of 655 nm LED only produces biased results. Not only PPFD level but also light quality may be considered for proper assessment of photosynthetic behaviours of the leaves in natural light. The authors dealt with the leaf angle applying the cosine law. However, when the light is incident on the leaf obliquely, reflectance increases, transmittance decreases, and the light path is lengthened. Thus, the situation is complicated.

Reply: We agree that leaf optics are complex and that both using values obtained using a meter that senses photons between 400 and 700 nm, and the simulation of that PAR are approximations of the true PAR. However, our previous paper, we demonstrated that the approach by showing that the Φ_{ii} values obtained using a fluorescence imager above a leaf were very similar to those measured upon clamping the device to the leaf, indicating that 1) the simulated PAR inside the clamp was a reasonable approximation of that reaching the leaf; and 2) at least over the time scale of interest the use of red rather than full spectrum light did not adversely impact the results.

We suggest that this result is due to the following: 1) the 655 nm light from the LED is absorbed evenly between PSII and PSI and antenna state transitions in vascular plants are quite small; 2) the time scale of the Light Potential experiments is much shorter than the time required for plant light quality responses, e.g. chloroplast movements, stomatal aperture changes and state transitions.

6) Also, this reviewer wonders the chamber volume and whether the air in the chamber was agitated. Stomatal responses may be neglected. However, if a leaf part is enclosed in a small volume of still air, CO₂ diffusion will be greatly suppressed.

Reply: This is an interesting point. The instrument's leaf clamp has channels to provide free air flow to the outside. To directly test this, we tested if adding additional airflow changed the results and found that it did not. We now directly address this in the text, as in the following:

Two features of the instrument's leaf clamp provide free air flow to the outside, preventing the depletion of CO₂ during the measurements. First, there is an approximately 3 mm space (or gap) between the leaf surface and the light guides, allowing lateral air flow. Second, this gap is connected to the external environment via a pair of rectangular (approximately 2 mm by 3 mm) air flow guides on the sides of the light guide, leading to the instrument case, the rear of which is open to the air. We thus do not expect any significant restriction of CO₂ diffusion to the leaf surface. To directly test for such effects, we compared measurements of ten separate mint leaves made with an unmodified instrument with ten made with an additional air pump that provided approximately 300 mL min⁻¹ air exchange in the leaf clamp. We observed no significant differences ($p > 0.5$) in LEF values measured as above, under ambient (LEF_{amb}) or high (LEF_{high}, 2000 $\mu\text{mol} \cdot \text{m}^{-2} \cdot \text{s}^{-1}$) red light, indicating that the airflow in the device was sufficient to prevent substantial CO₂ restriction to the leaves.

7) Light potentials: The authors use this phrase in the title as well as in the abstract. Because this term is not used widely, a solid definition is needed.

Reply: We appreciate the comment and have added the following into the Introduction:

We define the term "Light Potential" (LP) as the plant to respond to sudden increases in PAR, either by using it for productive photochemistry or up-regulating photoprotective mechanisms. In effect, LP is the response of photosynthesis to removal of light limitations.

Minor points

8) P. 19, 2nd para starting with A major outcome...: Is not it possible to analyze the present data with the meteorological data of the stand measured at the same time? For example, behaviours of two leaves under the same irradiance may be different depending on the environmental pre-histories of these leaves before the MultispeQ measurements.

Reply. We agree that this would be an interesting extension of the work, and could answer some interesting questions. In the current set of experiments, though, we did not compare multiple measurements on specific leaves, but rather sampled a wide range of different leaves. Comparing repeated measurements on the same leaves would require a different type of experimental setup and statistical analysis.

9) P.19, 4 th para: Unbiased... is OK, but it may be worth comparing the top and bottom leaves (although subjective).

Reply: This is an interesting comment. In this experiment, we did not record the leaf position in the canopy, partly because the canopy in mint was very deep. We have done something along these lines in both our first publication on MultispeQ (Kuhlgert et al. 2016) the results do suggest some interesting interactions between leaf position and developmental stages. We now mention these in the Results in the Discussion, see first paragraph on page 8:

We note that previous results, e.g., Kuhlgert et al. [33], indicate that there may also be significant interactions between canopy position and photosynthetic parameters, though the current experiment did not explicitly record these positions, but rather sampled them as described in Materials and Methods.

P. 25: Model 1 will be further unlikely if the authors take account of the presence of FR light in nature.

Reply: We think this is a very interesting and open question: Does the natural gradient of light quality in a canopy impact the over-reduction of PSI. Though our current data cannot resolve this, it could in principle be addressed in future work with the current instrument using the onboard 730 nm LED as a supplementary actinic source.

10) P.27: $\Delta\Psi$ is spontaneous while ΔpH is certainly slower. But, in high light it is a matter of a few second or so. More quantitative description is possible here.

Reply: We have added a short description of the kinetics of interconversion of $\Delta\Psi$ is spontaneous while ΔpH .

It is known that, initially after an abrupt increase in PAR, increased thylakoid pmf is stored as $\Delta\psi$; conversion of $\Delta\psi$ to ΔpH is controlled by the movement of counterions across the thylakoid membrane and protonation of lumenal proton buffering groups, occurs over the seconds to tens of seconds time scale (Cruz et al. 2001; Armbruster et al. 2016; Li et al. 2021; Zaks et al. 2012), and is dependent on the activities of various thylakoid ion transporters (Armbruster et al. 2014; Kunz et al. 2014; Davis et al. 2017). However, little is known about the natural diversity of $\Delta\psi/\Delta\text{pH}$ balancing.

11) The word 'data' is plural.

Reply: We thank the reviewer for pointing this out. We have changed the following:

pg. 8-- "Data from the PhotosynQ platform were reprocessed and cleaned..."

pg.8, line 4-- "However, all original data were maintained..."

pg. 29-- "The current data do not allow..."

pg. 31--" Data accessibility. Primary data are available on the photosynq.org site..."

Reviewer: 2

Comments to the Author(s)

The paper entitled "Light Potentials of Photosynthetic Energy Storage in the Field: What limits the ability to use or dissipate rapidly increased light energy?" by Kanazawa and coworkers (RSOS-211102) describes an analysis a set of physiological parameters, testing different properties/responses of the photosynthetic apparatus, gathered over a large sample of plants in the field, hence subjected to environmental fluctuations.

In particular the authors make use of device allowing to monitor different parameters by optical methods, either employing fluorescence or absorption transient, which are substantially well-established proxies for the study of photosynthetic responses. Using this set of information gathered almost simultaneously on each sampled leaf, it is then possible to asses limitations arising, principally, from limitation of at the level of Photosystem I, Photosystem II or intermediate component in the electron transfer chain, as well a possible synergies/compensatory effects of these factors. Together with the physiological parameters the intensity of incident light ("PAR radiation") and the leaves temperature are measured as well, allowing thus to statistically

correlated the physiological responses with the environmental status (at the time of the measurement).

To this end the author employ a non-supervised statistical algorithm capable of resolving correlations between the “environmental parameters, PAR intensity and leaves temperatures) for each of the physiological parameters estimated, and to distinguish, therefore, which of environmental condition is dominant and if an how, there is any inter-dependence (cross-correlation).

Based on the “grouping/sorting” of specific responses, it emerges that the factors which appears to limit photosynthesis in the field, i.e. for organism growth under non-controlled conditions, despite display a relatively large variability, as it may be expected, also show some distinctive features, which are particularly pronounced for low-PAR leaves (more often, but not always, corresponding also to low leaf temperatures). Under these conditions leaves in general appears to have a sub-optimal electron transport capacity, even when exposed to saturating intensity at the sampled area, and to have a strongly suppressed ability to develop regulative non-photochemical quenching at PSII level. Yet, in most conditions, for the organism tested (Mint) photosynthetic electron transport appears to be limited by the plastoquinone shuttle oxido-reduction, hence mainly by PSII/Cytb6f turnover, whereas PSI is found to be substantially non-limiting.

The study is interesting in that it explores the possibility to determine photosynthetic-limiting factors under “ambient growth” conditions, henceforth eliminating possible bias brought about by specific “laboratory/greenhouse” growth conditions. Nonetheless there are a few points that would need clarification and which are listed in the below.

Reviewer 2 questions:

1. One general comment is that in the paper, it is not specified how large the sample (number of leaves/independent plants) shall be in order for the statistical approach employed to be robust. This is a key information if the method is indeed to be transferred in practice for phenotyping/lines/cultivar or even species selection in a given environment. One suggestion I could advance would be that of a “jack-knife” like approach, i.e. excluding an increasing percentage of data from the sample (randomly sorted), and the very whether the clustering converges or not, within reasonable margin of confidence.

Reply: We thank the reviewer for this comment. Indeed, this is an interesting and emerging problem in the area of applied statistical research where the goal is to find the number of samples required to identify clusters in a complex multi-response and multivariate data. As discussed in more detail in the following (**More Detailed Reply, below**), although we cannot determine the

precise number of measurements needed to obtain useful results, we have taken two approaches to test the robustness of the clustering outcomes, one using cross-validations, and the other (which more intuitively relates the results to the interpretation) based on re-analyses of randomly selected sub-populations of the data set. Both show that the Clustering provided internally consistent results even with considerably smaller sample sizes, and thus the overall approach was robust. For reference, we have included data from the second approach in the manuscript in SI Fig. S14.

We have added the following to the main text (page 21, second paragraph):

As a first order test of the robustness of these clusters by re-analyzing randomly selected subpopulations of the data. As discussed in the legend to SI Fig. S14, we obtained comparable results, i.e. that we would interpret in similar ways, with as few subpopulations as small as 25% of the full data set, suggesting that the clustering approach was reasonably robust.

We also added additional text in the legend to SI Fig. S14.

More Detailed Reply: The GMM algorithm is capable of reducing model complexity and delivering robust estimations of the number and properties of the clusters, but there is not a straightforward test for how many data points would be needed to arrive at similar answers and a full exploration of the problem is probably beyond the scope of this paper. However, as we discuss below, it is possible to provide a first-order estimate of the robustness of the interpretations, or how many points are needed to obtain results consistent with particular interpretations.

There are several points to consider:

1) Because the approach is unsupervised one cannot know a priori what the identified clusters will represent, nor how many clusters will emerge. Thus, clustering with different subsets of points may lead to different cluster shapes or behaviors. In our case, though, we can use the reproducibility of the interpretable patterns contained in the clusters to determine if a smaller subset of data is a good representation of the whole.

2) The resolution of clusters into interpretable categories will depend on a number of factors, including: a) sample size; b) model complexity; and c) the granularity of the behaviors exhibited by the population (i.e., whether individuals exhibit clearly distinct behaviors or whether they can also show intermediate behaviours, see below). In the current case, it seems apparent that intermediate situations do occur, e.g., when both NPQ and photosynthetic control (PCON) contribute to modulating the light potential of linear electron flow (LEF). Thus, while more

granular behaviors will tend to fall into clusters that can be “safely” attributable to particular modes of action, the intermediate behaviors may variably fall into one cluster or another, depending on sample size, noise levels and other factors.

3) The problem is non-convex, and thus we applied an Expectation Maximization algorithm to estimate the parameters (Ref: Dempster, A.P., Laird, N.M., Rubin, D.B., 1977. Maximum likelihood from incomplete data via the EM algorithm. *J. R. Stat. Soc. B* 39 (1), 1–38.) We took into consideration the issue of model complexity by putting a penalty on the number of unknown parameters. Thus, while the algorithm is itself capable of reducing model complexity and delivering robust estimation, the required sample size estimation will be dependent on the number of unknown parameters to be estimated from the data.

4) We used a Gaussian Mixture Model (GMM) approach where all the samples (data points) are assumed to be generated from a mixture of multivariate Gaussian distribution with some unknown mean and unknown variance. The parameters estimated over here are the number of Gaussian mixtures (K), mean (μ_i) and variance (σ^2_i) for each cluster ($i=1,2,\dots,K$). (Ref: McLachlan GJ, Peel D (2000). *Finite Mixture Models*. John Wiley & Sons. doi:10.1002/0471721182.) For suitable model performance, Formann (1984) used a 2^D (preferably $5 \cdot 2^D$) many samples to be included where D is the number of model parameters to be estimated. Also, with an application of market segmentation study, Qiu and Joe (2009) suggested that the sample size should amount to a minimum of 10 times the number of variables in the segmentation base times the number of clusters. In the existing literature this approach seemed to work fairly well, for example, Kim et al. (2014) used model based clustering on hydrochemical data where a sample size of 102 was used to estimate two clusters, which is equivalent to estimate 5 parameters. In our data, 1054 samples maintained the criteria as well.

Despite these issues, it is possible to assess the impact of sample size on the conclusions we can draw from the clustering approach. In the following, we illustrate two ways to assess the robustness of the clustering interpretations.

Approach #1: Cross validation. To address the statistical model consistency, we performed further analysis on selected data using a five-fold cross validation approach. Because we are dealing with an unsupervised setup, we can not impose further assumptions, but one effective approach is to perform cross validation to test for breakdowns in "internal validity".

We illustrate using two data sets in the paper, Case 1, comparing NPQ_{amb} , PAR_{amb} and leaf temperature (T_{leaf}) (SI Fig. S5), and Case 2, comparing LEF_{High} , PAR_{amb} and T_{leaf} (SI Fig. S4), which differed in the extents of overlaps of the clusters along the PAR_{amb} axis. We randomly split the data into five approximately equal sized sub-groups (called folds). One fold was treated as a validation set, and GMM was performed on the remaining four folds. This process was

repeated five times so that each fold was used (in sequence) as a validation set. We then evaluated the average true positive rates (sensitivity scores) and the misclassification rates (or error rates). For both examples, we found average sensitivity scores greater than 85 percent and the average misclassification rates less than 20 percent.

It is noteworthy that these results include changes in classifications of Cluster 4 in Fig. 8, which spanned the two mechanistic models and likely represent intermediate behaviors, as well as between Clusters 1 and 2, which both fall into our mechanistic Model 3, and thus from a functional point of view may be considered to be redundant.

Approach #2: Effects of subsampling on robustness of mechanistic interpretations. In this approach, we assessed the robustness of internal validity by asking: To what extent can we decrease the sample size and still obtain clusters that are consistent with the given domain knowledge-based interpretation. Note that these domain model attributions were not dependent on the clustering itself. This approach assesses “internal validity”, i.e., whether a model is self-consistent. In this case, we are asking whether a smaller subsample gives us a similar (internally consistent) interpretation or conclusion.

The results in Fig. 8 showed four clusters that fell into either Model 2 or Model 3. Cluster 3 fell almost exclusively in Model 2. Clusters 1 and 2 fell almost exclusively in Model 3. Cluster 4, spans both Models 2 and 3, probably reflecting intermediate situations where both models partially apply. As discussed above, such behavior might reflect intermediate behaviors of the biological system. One might expect that Clusters 1,2 and 3 should be relatively robust indicators of Model #, and thus not be very dependent on how we subselect points whereas those in the “intermediate ” Cluster 4 may exchange between models.

Next, we sub-selected different subpopulations, consisting of 5, 25, 50 and 75% of the entire data set, randomly and re-performed GMM to ask the following:

1. What fraction of the time do we get clusters like Clusters 1, 2 and 3?
2. How consistently do they contain the “safe” points?
3. Do the points currently in cluster 4 move more frequently between the other clusters?

As expected, decreasing the number of sub-selected points changed the number of clusters output by GMM, and altered to some degree the shapes of the clusters. However, selecting 25% or more of the data leads to similar conclusions about the behaviour of photosynthesis within the clusters. In each case, one cluster fell predominantly within Model 2, a set (two or three) of clusters fell predominantly into Model 3, and one Cluster spanned both Models 2 and 3. Thus, if we combine the multiple clusters that fell exclusively into the Model 3 region, the approach appears to be robust with subsampling down to about 25% of the total data points. By contrast, clustering of only 5% of the data did not adequately reproduce the correlations with the mechanistic models, only two clusters appeared, one largely (but not exclusively) in Model 3 and the other spanning Models 2 and 3. Overall, although we cannot determine the precise number of measurements needed to obtain useful results, the methods described above show that the Clustering provided internally consistent results even with considerably smaller sample sizes, and thus the approach was robust. For reference, we have included this second approach in the manuscript in SI Fig. S14.

2- A point which appears to have been overlooked is the influence of “incident light spectrum” on top of the PAR intensity per se. especially for plants which tend to form rather dense

canopies, and Mint appears to do so, low intensity might be not only the result of clouding or other “sky” shading factors, rather because of leave-mediated filtering. Yet, in this case, the colour of the incident would also be much different than that of direct exposure, such as the canopy top. I would concede it may not be possible to check this parameter at posteriori. Nonetheless, the possibility that the PAR spectrum might play a role in the observed behaviour, shall at least be mentioned, since the largest limitations appears to have been determine at low-PAR when filtering is predictably more pronounced too. This might as well be correlated to leaf temperature too.

Reply: Please see response to the comment by Reviewer 1. We think it is an interesting idea that could be pursued in a future project, but would require new instrument capabilities to measure the far red and reproduce the observed changes.

3- I would not agree that the fast reversible qE component is always the dominant parameters in the data observed. Although it is certainly true that the authors observe a large rapidly reversible quenching fraction (that is assignable to qE), for large induction of NPQ (values reacting 3-4), and upon a relaxation of 2 NPQ units, it means the leaves are still significantly quenched to NPQ equivalents of 1-to-2. This is a 50%-75% excited state quenching, hence a large quenching. In terms of excited state the relaxed part, is, actually, much less. To better evaluate the effect of excited state quenching (rather than the quencher concentration, to which the “NPQ” parameter is related), it might far better to use the “yield of quenching” parameters ($=1-F_m'/F_m$), this is constrained to range from 0-to-1 (being a yield).

Reply: We have two (related) replies to this interesting comment, one on the effects of NPQ on LEF and the second on its effects on photoprotection. First, one way to interpret the question is: under what conditions does NPQ substantially control LEF. Somewhat surprisingly, in many cases, NPQ does not strongly (or even measurably) impact LEF, but does alter the redox state of PSII. This can be seen from results on the Arabidopsis *npq4* mutant which is deficient in the qE response and thus has a smaller extent of NPQ under most conditions. The responses of LEF (or Φ_{ii}) to photosynthetically active radiation (PAR) are similar between *npq4* and the wild type, but the source of limitations is different, with the wild type showing stronger limitations caused by dissipation of absorbed light energy by NPQ, whereas in the mutant the limitation is mainly from accumulation of reduced Q_A , which likely promotes photodamage. The reason for this effect is that PQH₂ oxidation at the cytochrome b6f complex remains the rate-limiting step in overall LEF until NPQ becomes quite high, though it does decrease the “excitation pressure” and associated accumulation of electrons on Q_A . What is exciting for us is that, in the work described in the manuscript, we see clear evidence that, under field conditions, qE does impact not only on Q_A redox state but also LEF.

Second, we completely agree that the impact of NPQ on photoprotection (or related processes such as the accumulation of electrons on Q_A) will also depend on the photochemical state of the photosystems so that the true impact of the rapidly and slowly relaxing forms of NPQ would depend on PSII redox state. Further, this competition will also be impacted by environmental conditions. The data set does include the Φ_{NPQ} parameter, but we did not include it in the current paper because its interpretation is more complex (since it is impacted by both photochemistry and NPQ) while the individual effects are already recorded by the individual NPQ and Φ_{ii} parameters. We now clarify in the text that we aim to describe the fraction of NPQ that was able to be rapidly adjusted, rather than the fraction of absorbed photons that are dissipated by NPQ processes. We modified the text to cover these issues, and in page 11, last paragraph:

Overall, these results indicate that a large fraction (in many cases the majority) of NPQ_{amb} as well NPQ_{high} recovered within 10s of darkness and can likely be attributed to qE, and thus, under our conditions, qE is likely to be the most important form of NPQ for rapid adjustments to photoprotection. The residual, more slowly reversible, components reaching a little above 2 are likely to include qI or qZ [45,46], although the limited time frame for the protocol does not allow us to rule out contributions from longer-lived qE. It is also important to consider that the fraction of light energy dissipated by the NPQ, i.e., Φ_{NPQ} , will also depend on the fraction of PSII in open states [37], which will also be impacted by ambient and fluctuating light, T_{leaf} and other factors.

Some specific points:

1- summary, line 35 “efficient light capture”, I trust this shall be “light conversion”.

Reply: We have modified the text to read: “...efficient conversion of light energy...”

2- introduction line 33-34 and in other places: “photo-protective capacity”. Whereas I fully agree with the statement concerning balancing of non-photochemical and photochemical quenching, the extent by which NPQ effectively protects PSII is yet to be unambiguously determined, with suggestion it could even be rather modest, unless quenching is very large. It would certainly act as a regulative process, in terms of electron transport, irrespective of the direct protective effect on light-induced damage.

Reply: As covered in our reply to the previous point, the impact of NPQ on photoprotection will occur at lower NPQ than that required to substantially slow electron transport. We do take care to emphasize that the connection to photoprotection is hypothetical.

3- page 10-11, often the terms “phenotype” appears, which, although understandable as a “functional” term, is a bit odd in this context, as it might refer even to different leaves within the same plant, and hence clash with the most common use of the term. I might suggest the term “response behaviours” or something alike.

Reply: We agree that the term “phenotype” can be confusing and have replaced it as suggested.

4- Figure 3, 4 and in general. It might be interesting to present, even in the supplementary information a direct correlation plot of the measured parameters, i.e. LEF vs NPQ, LEF vs ECS, etc.

Reply: We completely agree that such plots are interesting! But, perhaps too interesting. The vast parameter space enclosed by such plots presents a big issue for a publication like this, in that each apparent relationship will require a detailed explanation. After many hours of explorations of this type, we decided that we should focus on presenting results that could test specific models. However, the entire data set is open and accessible from the PhotosynQ.org web site, which has the tools needed for anyone to further explore the results.

5- Page 19, line 19 “rapidly variable”, I trust this shall be “reversible”

Reply: The nuance we were attempting to convey was that the NPQ was rapidly formed and reversed, thus we chose the term ‘variable’. However, the reviewer sensed that this could be an unusual usage, so we have replaced ‘variable’ with ‘reversible’.

6- Page 22, paragraph from line 14. The statement seems broadly unjustified. It is clear, even from clustering, that there is a correlation between the environmental factors and the measured parameters. In case of controlled environment the response measured shall still be correct, but refer preferentially, to the specific growth conditions employed. Furthermore later on the authors themselves stated that further investigation, possibly by controlling the conditions (presumably to limit the number of stochastic “environmental” parameter) shall be needed to better understand the trends observed in this paper. It would be fairer to say that the comparison of “environmental fluctuations” (not all of which could be monitor) and, possibly a set of “controlled conditions” (rather than just “a” controlled condition), would be more informative in the future.

Reply: The concept we were trying to convey is that measurements in the field following a sudden perturbation of the light intensity, for instance increasing the light to full saturation, will not directly reflect the state of the plant prior to that. To clarify, we have modified the text by specifying that measurements following light changes can be informative, but care must be taken when extrapolating back to the non-perturbed, natural conditions, as in the following (See page 17, second paragraph in revised version):

These results also imply that accurate estimates of LEF, NPQ and other photosynthetic parameters *under natural conditions* will require measurements under ambient light, because sudden changes in PAR can lead to severe perturbations in photosynthetic limitations or regulation. Attempts to “simplify” field experiments by setting PAR to some constant value will lead to strong perturbations and the measured values will reflect these perturbations. The effects are vividly demonstrated by the opposite dependencies of $\Phi_{2_{amb}}$ and $\Phi_{2_{high}}$ on PAR_{amb} (Fig. 2D), and validate the use of the PAR matching feature of the MultispeQ instrument. Nevertheless, as shown here, the effects of these perturbations can be informative, but care must be taken in extrapolating to the non-perturbed state. It is also important to keep in mind that the rates of acclimation may vary substantially between species, and that these may be assessed by performing more intensive experiments with variable high light and dark recovery times.

7- same page, line 37, as in the general comments. The majority of quencher appears to be of the qE type, the majority of the quenching, as decrease in excited state population, not necessarily so, depending on the individual sample. Residual, slowly reversible/irreversible quenching, delta-pH unrelated, appears to be dominant, in terms of dissipation in a good fraction of the sampled leaves.

Reply: We did not intend to imply that slow relaxing forms do not contribute, but that under most conditions, the largest fraction of NPQt was rapidly reversible. We were somewhat surprised by this finding and felt it should be mentioned. We have modified the text to better reflect this (see page 17, 3rd paragraph in revised manuscript): “The rapid reversal of NPQ_{amb} and NPQ_{high} over 10 s of dark indicated that, under our conditions, a large fraction of NPQ is in the form of qE (Fig. 3B and C)...”

8- Page 26 line 14, and following. “net reduction/net oxidation”, etc. This is with respect to no-regulative process taking places.

Reply: Thanks for pointing out the ambiguity. We have clarified this in the text (page 20, second paragraph in revised manuscript), as in: “The expected theoretical changes in measurable parameters upon activation of the three models are indicated by the coloured boxes in the figure, and can be related to Models 1-3 in Scheme 1...”

9- page 20. Same as point 6

10- Conclusion. I'd recommend to specify, what Model 1/2/3 are here, i.e. (control by qE, by photosynthetic control, PSI donor/acceptor limitation.

Reply: We agree and have modified the paragraph to more clearly define the models (see page 23, last paragraph):

Such combinations of rapid measurements allowed us to test for various models over broad-scales by looking for internally-consistent relationships among the various measured parameters. For example, we observed no evidence for Model 1 (limitation at the acceptor side of PSI) behaviour in the current study, but we do not exclude the possibility in different species and/or different environmental conditions. The analysis supports the operation of Models 2, the rapid activation of NPQ resulting in net oxidation of Q_A^- , and 3, the strong activation of PCON, resulting in accumulation of Q_A^- . We surmised that Model 2 behavior would be the most photoprotective, while Model 3 type behaviour would likely lead to photodamage, though we do not have independent endpoint measurements (e.g., yield, growth rates etc.) to validate that the propensity for Model 3 behaviour has long-term consequences. Further, The models are not exclusive, and there will almost certainly be cases, e.g., Cluster 4 in Fig. 8, where intermediate behaviours will be apparent, either because of co-limitations among multiple processes or heterogeneity between chloroplasts in the leaf samples.

11- page 28: “upcoming paper”, sound somewhat mysterious for an “open science” journal. Details on how the presented data and methods are effectively intended to be made broadly accessible shall be specified.

Reply: We agree and have removed the phrase.

12- the authors often refer to the protocol employed with “quasi-commercial” acronyms. Yet, since the protocols and approaches rely on the quantities measured, which, provided estimated correctly, can be assessed by different commercial or laboratory-developed instruments, there's no need or reason, to refer to "specific" commercial/jargon name for them.

Reply: There is no intention of making the names “commercial”, but only to define them as distinct. We have found this to be important because there are so many variations of measurements in the field and re-describing when used complicates the text. Such nomenclatures are common in fields such as EPR (e.g., ESEEM) or mass spectroscopy (e.g., MALDI TOF).

13- I find it a shame that most of the parameter clustering is presented in the supplementary material only, as it is informative and successive analysis rely on it. Nonetheless is accessible.

Reply: We are grateful for this comment. We chose to limit the clustering in the Main text to avoid overwhelming the readers. We are also writing up a more detailed manuscript targeted to a statistics journal, which will highlight these types of analyses in more detail.

Yet, I have a slight concern. Whereas some of cross-correlation are very clear, some other, albeit appears to be significant (according exactly to which significance test, since it is not specified?), might be in part biased by a few outliers (e.g cluster 6 figure S3 bottom panels of the matrix, S4, bottom corner, and so on).

Reply: We appreciate the point and have tested for the effects of “influential” outliers. After the estimation of clusters, we have looked into each cluster and exploited the linear dependence of different traits in those clusters. We have used the significance test of correlation coefficient using *cor.test()* function in R, where the p value is calculated using a t-distribution with $n_i - 2$ degrees of freedom, where n_i is the number of samples in a given cluster. (reference: Casella, George, and Roger L. Berger. 2002. *Statistical inference*. Belmont, CA: Duxbury.).

While some data points may look like outliers, they may not be “influential” since they are closer to the regression surface (Draper, N. R., & John, J. A. (1981). Influential Observations and Outliers in Regression. *Technometrics*, 23(1), 21–26). For model diagnosis, one of the most common measures of influence is Cook’s distance which is a function of both leverage and standardized residuals. A Cook’s distance (CK_i) is considered to be large if for an observation i , $CK_i > 4/n$ and heuristically, an observation with a large Cook’s Distance is called influential (Reference: Cook, R. Dennis (March 1979). "Influential Observations in Linear Regression". *Journal of the American Statistical Association*. 74 (365): 169–174.) For our supporting information figures, we have calculated the Cook’s distance for different observations using the R function *cooks.distance()* and found that the observed outliers are *passive* and rather than *influential* outliers. For example, for Cluster 6 in SI Fig. S3, we have plotted points without the *passive* outliers and show that the statistical significance of the correlation coefficient(s), presented in the main text remains valid.

Cluster 6

Modified correlation matrices between LEF_{amb}, PAR_{amb} and leaf temperature (T_{leaf}) for Cluster 6 from SI Fig. S3. The different subplots represent the co-dependence between the parameters used. For each subplot the diagonal entries are the density plots for individual parameters, lower triangular entries represent the plots taking two parameters at a time and upper triangular entries are the Pearson correlation coefficients for the selected parameters.

Also, it is noteworthy that the clustering was done on a multivariate space, so the *passive* outliers may appear to occur in our two-dimensional representation, but may not be *actual* outliers in higher dimensional space. For example, in Cluster 6 of SI Fig. S3, passive outliers seem apparent in the space of PAR_{amb} and T_{leaf}, but not in the space of PAR_{amb} and LEF_{amb}.

Reviewer: 3

Comments to the Author(s)

The title and subject of the manuscript are very interesting from a methodological and practical point of view, suitable and adequate. The scientific content contributes to the space in which it develops. The work was provided with a sufficient level of scientific novelty.

This seems to be well-conducted research and to have clear, repeatable results. The authors studied the responses of plant photosynthesis to rapid fluctuations in environmental conditions. Authors suggest With decreasing leaf temperature or PAR, limitations to photosynthesis during high light fluctuations shifted from rapidly-induced NPQ to photosynthetic control (PCON) of electron flow at the cytochrome b6 f complex. At low temperatures, high light induced lumen

acidification, but did not induce NPQ, leading to accumulation of reduced electron transfer intermediates, a situation likely to induce photodamage, and represents a potential target for improving the efficiency and robustness of photosynthesis.

I have no critical comments. Paper is very innovative. The structure of the paper is logical and the results are well reproduced. The introduction and discussion are well organized. Results reported have not been published elsewhere. Conclusions are presented in an appropriate fashion and are supported by the data. I think the overall concept is interesting and potentially important.

Reply: The authors thank the reviewer's positive comments.

I suggest reading/include some references of W. Huang, Ch. Miyake, Takagi, L. Ferroni.

We agree and have modified the text to include additional background (with additional references, including to all the authors mentioned) on the effects of fluctuating light on photosynthesis.

I recommend to ACCEPT the paper for publication.

Reply: Thank you!

Appendix B

Responses to Reviewer Comments (RSOS-211102.R1): Light Potentials of Photosynthetic Energy Storage in the Field: What limits the ability to use or dissipate rapidly increased light energy?

We are very grateful for the insightful and positive comments from you and the reviewers and happy that the paper can be accepted with minor revisions.

We have addressed the remaining issues brought up by Reviewer 2, most importantly adding clarifying text to address the question of residual NPQ on photochemistry. We also discuss and support our use of specific nomenclature as needed to properly identify the parameters and methods we use.

We did find an issue with one point brought up by Reviewer 2, where the definition of ϕ NPQ (the quantum yield of nonphotochemical quenching) was defined so that it would not apply to the conditions of photosynthesis seen in a field. We added a clarifying sentence to the text to ensure that the reader will not be confused.

We again thank the Reviewers and Editor.

Dave

Reviewer comments to Author:

Reviewer: 1

The revised version reads well.

Reply: Thank you!

Reviewer: 2

Either in the revised version of the paper entitled “Light Potentials of Photosynthetic Energy Storage in the Field: What limits the ability to use or dissipate rapidly increased light energy?” (RSOS-211102.R1) or through the point-by-point answer to reviewer comments, the author have addressed carefully and thoughtfully the majority of the point raised.

Although I would therefore consider the paper recommendable for publication, I would also like to raise two minor points, already signalled in the previous comments, that I trust can be addressed pretty straightforwardly by the authors.

Original point 12- In the answer the author states "...there are so many variations of measurements in the field and re-describing when used complicates the text. Such nomenclatures are common in fields such as EPR (e.g., ESEEM) or mass spectroscopy (e.g., MALDI TOF)."

I trust the author had no intention of making "commercials" for the instrument/methods. Yet, what they are discussing is a combined analysis from different experimental approaches and does not related to the acronyms used as counter-example which are proper spectroscopic methods (i.e. ESEEM, ENDOR, DEER, etc.. in EPR are specific MW/RF pulse sequences to investigate electron-nuclear or electron-electron interactions; same as in mass spectroscopy). What's discussed/proposed here is more of combination of methods, not simultaneously, but sequentially applied to the material, and that can be (and likely will) further expended, instrumental flexibility allowing.

Reply: We have looked back at the original point, in which the reviewer stated: "12- the authors often refer to the protocol employed with "quasi-commercial" acronyms. Yet since the protocols and approached rely on the quantites measured, which, provided estimated correctly, can be assessed by different commercial of laboratory-developed instruments, there's no need or reason, to refer to "specific" commercial/jargon name for them."

Reply: It is not clear to us which specific nomenclature was thought to be "quasi-commercial" or not needed. We feel the nomenclature we used is appropriate and required to properly identify and distinguish the parameters we use, especially when the methods should be distinguished from those of precious works. Below, we summarize our nomenclature and our reasoning for the inclusion of each term or set of terms:

- *We introduced the term "Light Potential" and gave it the abbreviation "LP". Reviewer 1 commented that this term was novel and requested a firmer definition, which was supplied in our revisions. We feel the concept is distinct and should be designated as such. The values measured after the light exposure tell us something distinct from those measured in the steady state. They tell us what the potential for increased photosynthesis or photoprotection would be if light limitations were rapidly removed.*
- *We also designate LEF and other parameters with subscripts, but this is needed to denote the conditions under which they were taken. In fact, we spent quite a lot of time exploring different ways of designating these*

parameters. Most of them were awkward or difficult to read. The approach we settled on has the benefits of compactness, but since the subscripts contain meaningful text, e.g. “amb” and “high” for ambient and high light, they are readable.

- *The remaining terms were already established in the literature some time ago to designate specific parameters and are in common use, beyond our laboratory. For example, terms for the versions of NPQ that we use (NPQt, qIt, qEt, qL) were established some time ago (Tietz et al. 2017; Kramer et al. 2004) and, as described in the text. It is important to distinguish the NPQt-related these from those parameters measured using a different approach because, although the values are qualitatively similar, the magnitudes are different. The electrochromic shift parameters (ECSt, gH+) were also introduced some time ago, and are described in a highly cited review on the methodologies (Baker, Harbinson, and Kramer 2007). For accuracy, we think it is important to maintain these.*
- *The terms GMM and Bayesian Information Criterion (BIC) are standard, commonly used nomenclature in fields of statistics.*
- *The term PCON is, indeed, uncommon, but because it is used frequently throughout the paper, we felt the text benefits from a distinct designation.*

- Original main point 3 – I trust the authors have only partially addressed the point raised. As correctly state, there are two issues, one concerns the relation between quenching and inhibition: that has properly addressed within the paper framework.

The second is the impact of fraction of reversible/non (rapidly) reversible NPQ on regulation in broad sense. The point raised was that is not so much the fraction of reversible NPQ, rather the extent of residual quenching (after a dark/low light) transition, that likely represent the key parameters. To give two limit scenarios, one can think a NPQ which develops to a level of 2, and fully recovers. The other in which develops to a level of 4 and recovers to a level of 2. In the first case the systems, after recovery, is fully unquenched and photochemical yield is therefore unaffected (this will be the case for a any “small” value of NPQ, since the rough equivalent for photochemical quenching is around 4-5). In the other case of large quenching development (for the same recovery), the system after the transition remains with a non-photochemical quencher which, to some extent, is in principle capable of competing (kinetically) with photochemistry instead.

I on the other hand fully disagree with the statement that the $\phi(\text{NPQ})$ parameter depends on photochemistry (or that it does differently from the NPQ one). If, as to my knowledge, NPQ is defined as $\text{NPQ} = F_m/F_m' - 1$ and $\phi(\text{NPQ}) = 1 - F_m'/F_m$ they contain exactly the same terms, all estimated at “closed centres” where photochemistry shall not be a factor. If it were to be, it would be for both parameters. It is just NPQ is proportional to the quencher(s) concentration(s) and $\phi(\text{NPQ})$ is a quenching “yield-like” parameters, varying from 0-1.

Reply: We agree that any NPQ that is not recovered after a light pulse will contribute to a loss of photochemical efficiency, and we did not intend to imply that it will not. We point out that the definition the reviewer gave for “ $\phi(\text{NPQ})$ ”, i.e., $\phi(\text{NPQ}) = 1 - F_m'/F_m$, can only be applied to special conditions (under super-saturating pulses of light, see below) and thus cannot be used to describe Φ_{NPQ} under normal illumination. To clarify both of these issues, we have added the following sentence to the Discussion (Page 17, third paragraph).

“It is important to note, though, that residual NPQ will contribute to decreases in photochemical efficiency, and that the extent of these effects will also be impacted by other factors, including the redox state of Q_A [37].”

In the following, we provide a more detailed reply, but note that this topic has been covered previously in the literature (e.g., Kramer et al. 2004) and thus we do not think it would be appropriate to expand the paper to cover the effects in any detail, but we are more than happy to add something to the SI if requested.

The derivation of Φ_{II} and Φ_{NPQ} is based on the standard Stern-Volmer model for competition between decay routes for absorbed excitons. The Reviewer’s version of the Φ_{NPQ} parameter gives the following:

$$\Phi_{\text{NPQ}} = 1 - \frac{F_m'}{F_m} = \frac{k_{\text{NPQ}}}{k_f + k_d + k_{\text{NPQ}}}$$

where k_f , k_d and k_{NPQ} are the rate constants for decay of exciton by fluorescence, basal (or unregulated) non-radiative processes and NPQ. This can represent the yield of NPQ, but only when all PSII centers are closed, i.e., in very special cases, e.g. when the leaf is exposed to short super-saturating pulses of light. Importantly, this version does not apply when PSII centers are (partially or fully) open, as is the case for plants in the field. In these cases, the appropriate definition of ϕ_{NPQ} for these cases should be:

$$\text{PhiNPQ} = \frac{k_{NPQ}}{k_f + k_d + k_{NPQ} + q_L \cdot k_{pc}}$$

where q_L is the fraction of Q_A in its oxidized state and k_{pc} is the intrinsic rate constant for photochemistry by open PSII centers, as described previously (Kramer et al. 2004). These additional terms are needed because under real conditions, photochemistry is non-zero. The distinction is important because the yields of photochemistry (Φ_{ii}) and NPQ $\text{phi}(\text{NPQ})$ will depend on each other, in non-linear ways because both the terms for NPQ and photochemistry appear in the denominator, with two important consequences: 1) phiNPQ will not be linearly related to the concentration of the NPQ quencher; and 2) the dependence of phiNPQ on the NPQ quencher will also depend on the rates of other quenchers, especially photochemistry, which in turn will be determined by the fraction of open PSII centers (or q_L). The important point is that the loss of efficiency will also depend on other factors, especially the fraction of PSII centers with oxidized QA, which will impact the effects of rapidly and slowly recovered NPQ.

We feel that the addition described above will cover this issue without unduly complicating the text.

- Baker, N., J. Harbinson, and D. M. Kramer. 2007. "Determining the Limitations and Regulation of Photosynthetic Energy Transduction in Leaves." *Plant, Cell & Environment* 30: 1107–25.
- Kramer, D. M., G. Johnson, O. Kiirats, and G. E. Edwards. 2004. "New Fluorescence Parameters for the Determination of Q_A Redox State and Excitation Energy Fluxes." *Photosynthesis Research* 79: 209–18.
- Tietz, Stefanie, Christopher C. Hall, Jeffrey A. Cruz, and David M. Kramer. 2017. "NPQ(T): A Chlorophyll Fluorescence Parameter for Rapid Estimation and Imaging of Non-Photochemical Quenching of Excitons in Photosystem II Associated Antenna Complexes." *Plant, Cell & Environment* 40 (8): 1243–55.